# Theory of Continual Learning Against Data Poisoning Attacks

**Yiting Hu** [1] [2]   **Lingjie Duan** [2]

## Abstract

Continual learning (CL), where a model is trained on a sequence of data tasks, is increasingly being adopted across key fields such as large language models and image recognition, yet it remains highly vulnerable to data poisoning that triggers learning divergence or severe excess risk. Despite these threats, a principled theoretical foundation in CL for understanding attack and defense remains lacking. In this paper, we develop a theoretical framework to analyze strategic attacks and defenses in regularization-based CL, a cornerstone of recent CL theory. By framing the adversary-defender interaction as an online zero-sum game, we first establish a fundamental performance limit: no defense succeeds when an adversary poisons a linear proportion of tasks by injecting unbounded noise or pattern shifts in regularization-based CL. We then analyze two possibly defensible scenarios: infrequent attacks and bounded noise per attack. For the former regime, we propose a task-to-task verification mechanism to detect data poisoning and reduce cumulative bias for learning convergence. For the latter regime, we derive a robust defense that minimizes the model's sensitivity to poisoned features, provably accelerating the convergence rate. Extensive experiments on realistic tasks further validate our theoretical results.

## 1. Introduction

Recently, continual learning (CL) has become prominent in machine learning as models increasingly need to learn from sequentially arriving data tasks. The major challenge in CL is managing the stability-plasticity trade-off: adapting to new data distributions while preventing the catastrophic forgetting of previous knowledge. Numerous algorithms employing empirical and experimental methods have been proposed to manage knowledge transfer (Kirkpatrick et al., 2017a; Aljundi et al., 2018; Liu & Liu, 2022; Chaudhry et al., 2019; Jin et al., 2021). To move beyond heuristic approaches, a growing body of literature focuses on regularization-based CL to provide rigorous generalization bounds (Peng et al., 2023; 2025). For example, Evron et al. (2022) and Lin et al. (2023) analyze forgetting loss in the minimum norm estimator for continual linear regression, while Zhao et al. (2024) analyze regularization-based CL, introducing a provably optimal regularization-based algorithm for continual linear regression. Another line of work analyzes CL in the Neural Tangent Kernel (NTK) regime, bridging between nonlinear and linear CL models (Bennani & Sugiyama, 2020; Doan et al., 2021). These studies underscore the potential of regularization-based CL to bridge the gap between theory and practice.

However, prior theoretical studies overlook potential security threats in CL. Recent literature provides empirical evidence through experimental evaluation that CL models are susceptible to various adversarial exploits. These experimental findings range from targeted attacks, such as backdoors designed to trigger specific misclassifications (Umer et al., 2020; Guo et al., 2025), to availability attacks that demonstrably degrade overall generalization performance via data poisoning. In particular, Han et al. (2023), Li & Ditzler (2022), and Abbasi et al. (2024) demonstrate how data poisoning can erase learned knowledge and induce catastrophic forgetting, revealing the inherent susceptibility of CL models. Nevertheless, a principled theoretical foundation remains elusive, leaving the underlying mechanisms governing adversarial dynamics in the CL regime largely uncharacterized.

A unique vulnerability of CL is the capacity for persistent adversarial manipulation throughout the learning trajectory. By targeting multiple tasks over time, an adversary can sequentially bias the model, steering its long-term convergence toward an adversarial goal. We illustrate these attacks in Fig. 1, where tasks $t$ and $t'$ are poisoned within a specific budget to prevent the convergence of excess risk.

To address these theoretical deficiencies, this paper examines data poisoning as a fundamental vector for availability

[1]Singapore University of Technology and Design, Singapore [2]The Hong Kong University of Science and Technology (Guangzhou), Guangzhou, China. Correspondence to: Lingjie Duan <lingjieduan@hkust-gz.edu.cn>.

*Proceedings of the 43rd International Conference on Machine Learning*, Seoul, South Korea. PMLR 306, 2026. Copyright 2026 by the author(s).

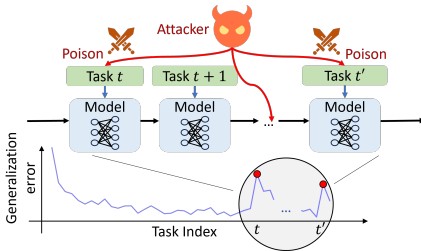

*Figure 1.* In CL, a new task arrives at each time step, and the model is trained on this task with the expectation of gradual convergence over time. Meanwhile, an adversary can poison specific tasks (e.g., at time $t$ and $t'$) in CL to boost the excess risk.

*Table 1.* Provable Defensibility of Data Poisoning Attacks in Continual Learning

| Data Poisoning Characteristics | | | Our Analysis Results |
| --- | --- | --- | --- |
| **Frequency** | **Magnitude** | **Pattern** | |
| Frequent | Bounded | Shifted | No provable convergence (Thm. 3.1) |
| Frequent | Unbounded | Shifted/Non-shifted | No provable convergence (Thm. 3.1) |
| Infrequent | Unbounded | Shifted/Non-shifted | Guaranteed convergence (Thm. 4.4) |
| Frequent | Bounded | Non-shifted | Fast convergence (Thm. 5.3) |

attacks. Given its capacity to globally degrade model performance, we leverage this attack class to derive a unified analytical framework for understanding adversarial resilience in CL. We formally characterize the adversarial design space in data poisoning across three pivotal dimensions, as summarized in Table 1, to develop a rigorous evaluation of attack dynamics: (i) the temporal frequency of poisoning across sequential tasks, (ii) the bounded/unbounded magnitude of the poisoning budget per attack, and (iii) the pattern of the poisoning (i.e., whether it induces a distributional shift).

To mitigate the impact of such attacks, regularization-based robust learning has emerged as a prominent paradigm for bolstering algorithmic resilience against noise and perturbations. For instance, Yan et al. (2018) integrated adversarial perturbation-based regularizers into classification objectives, while Li et al. (2021) applied regularization on embedding spaces. Motivated by the empirical efficacy of regularization-based methods against attacks and the recent theoretical shift toward such strategies in CL, we ground our defense framework within this paradigm, allowing us to leverage its analytical tractability to establish the first formal foundations for adversarial resilience in the CL regime.

Another significant class of CL algorithms comprises memorization-based methods (Hurley et al., 2009; Rebuffi et al., 2017; Frederickson et al., 2018; Kim et al., 2021; Wang et al., 2023). However, these approaches are often heuristic compared to regularization-based paradigms, rendering the derivation of a unified theoretical framework analytically arduous. Consequently, memorization-based methods serve primarily as empirical benchmarks for comparison with the theoretical results of this study.

We aim to answer two key questions in this paper:

*(i) To what extent do there exist fundamental limits where data poisoning attacks are provably indefensible within the CL regime?*

*(ii) Within feasible defense regimes, how can regularization-based CL objectives be optimized to ensure convergence?*

Our novelty and contributions are summarized as follows.

- *New theory of robust CL against data poisoning attacks*: We formulate the adversary and system's attack-defense problem as an online zero-sum game, and systematically characterize the adversary's capability in Section 2. In Section 3, we establish a fundamental performance limit, proving that no provable convergence guarantee can be established if the adversary poisons a linear fraction of tasks via unbounded noise or data pattern shifts.

- *Task-to-task verification defense against infrequent and unbounded attacks*: Section 4 explores a defense regime against finite sequences of unbounded adversarial attacks. To mitigate cumulative bias along the learning trajectory, we propose a Task-to-Task (T2T) verification mechanism. By decoupling historical updates from the current model state, T2T identifies attacks in recent tasks without being affected by the propagation of adversarial influence.

- *Robust feature defense against frequent, bounded and non-shifted attacks*: Section 5 investigates widespread, bounded, non-shifted attacks, where the adversary strategically allocates its poisoning budget to inject noise and maximally degrade convergence. To address this, we solve the online zero-sum game to derive optimal regularization parameters, yielding a robust defense that mitigates feature sensitivity to attacks and accelerates the convergence rate.

- *Experiments on realistic datasets*: In Section 6, we conduct empirical evaluations on real-world datasets to validate our theoretical insights. We compare our approach against existing regularization and memorization-based methods. The experimental results closely align with our theoretical findings and demonstrate superior performance over existing baselines.

## 2. System Model and Problem Formulation

In this section, we begin by introducing the CL model. We then formulate the adversary-defender framework and model the adversary's data poisoning attacks along three key dimensions: frequency, magnitude and pattern. Finally,

we present our preliminary formulation of the tractable adversary's problem within the regularization-based defense framework, to be further developed in subsequent sections.

## 2.1. Continual Learning Model

In a standard CL setup, we encounter a sequence of tasks indexed by $t = 1, \ldots, T$, arriving sequentially. Each task $t$ is associated with a dataset $D_t = \{(x_t^i, y_t^i) \in \mathcal{X} \times \mathcal{Y}\}_{i=1}^{n_t}$, where $n_t$ represents the sample size for task $t$. $\mathcal{X} \subseteq \mathbb{R}^{p_1}$ is the feature space and $\mathcal{Y} \subseteq \mathbb{R}^{p_2}$ is the label space. We assume that all data samples in $D_t$ are drawn from a distribution $\mathcal{D}_t$. The distribution $\mathcal{D}_t$ may remain stationary or may vary across different tasks $t$, where the latter case also corresponds to domain-incremental learning (Van de Ven & Tolias, 2019). The objective in CL is to learn a global model $f_w : \mathcal{X} \to \mathcal{Y}$, parameterized by $w \in \mathbb{R}^p$, that minimizes the global excess risk across all $T$ tasks:

$$\mathcal{R}_T(w) = \sum_{t=1}^{T} \mathbb{E}_{z \sim \mathcal{D}_t}[\ell(w, z)] - \min_{w^*} \sum_{t=1}^{T} \mathbb{E}_{z \sim \mathcal{D}_t}[\ell(w^*, z)], \tag{1}$$

where $\ell(w, z)$ is a loss function that quantifies the discrepancy between the model's prediction and the ground-truth label $y$, with $z = (x, y)$.

Any regularization-based strategy can be formulated as the following update under a generalized $\ell_2$-regularization framework, which has been proven to be equivalent to a broad class of methods, including the minimum-norm estimator and ridge regression (Zhao et al., 2024). At any time $t \in [T]$:

$$w_t \in \arg\min_{w'} \left\{ \frac{1}{n_t} \sum_{z \in D_t} \ell(w', z) + \frac{1}{2} \|w' - w_{t-1}\|_{H_t}^2 \right\}. \tag{2}$$

In this framework, the first term on the right-hand side of $w_t$ above quantifies the estimation error of the updated model on the current task $t$, while the second term controls the update distance from the previous model $w_{t-1}$ using the regularization matrix $H_t \in \mathbb{R}^{p \times p}$.

## 2.2. Data Poisoning Attack Model in CL

We assume training-phase poisoning attacks by a strong adversary who knows the current CL model state and dataset to deploy targeted and strategically crafted perturbations.

When the adversary targets task $t$, it introduces a perturbation $\eta_t = (\eta_{x,t}, \eta_{y,t})$ into the clean dataset $D_t$, subject to the poisoning budget constraint $\mathbb{E}[\|\eta_t\|_F^2] \leq M$. As a result, the system receives a poisoned training dataset upon each attack:

$$D_t = \{(x_t^i + \eta_{x,t}^i, y_t^i + \eta_{y,t}^i) \in \mathcal{X} \times \mathcal{Y}\}. \tag{3}$$

We can further decompose the perturbation as $\eta_t = \hat{\eta}_t + \tilde{\eta}_t$, where $\hat{\eta}_t$ represents the conditional mean shift, and $\tilde{\eta}_t$ denotes a non-shifted stochastic component satisfying $\mathbb{E}[\tilde{\eta}_t \mid \mathcal{I}_t] = 0$, with $\mathcal{I}_t$ denoting the information available when the perturbation is chosen, including the history and the current dataset.

We characterize the adversary's capability along three dimensions: (1) the number of data tasks $K$ the adversary can poison over time, (2) the magnitude of the poisoning budget $M$ per attack, and (3) the poisoning pattern in each attack (to have the shifted part $\hat{\eta}_t$ or not). Specifically, we classify adversarial capabilities as follows:

**Definition 2.1.** *Frequent or infrequent attack type*: an attack is frequent if $K$ scales linearly with time, i.e., $K = \Theta(T)$. Otherwise, an attack is infrequent when $K = O(1)$, meaning the adversary targets only a finite number of tasks regardless of the total learning horizon $T$. [1]

**Definition 2.2.** *Unbounded or bounded attack type*: an attack is unbounded if $M$ scales linearly or superlinearly with time, i.e., $M = \Omega(T)$. Otherwise, an attack is bounded when $M = o(T)$, meaning $M$ scales sublinearly relative to $T$.

**Definition 2.3.** *Shifted or non-shifted attack type*: an attack is shifted if the shifted component $\hat{\eta}_t \neq \mathbf{0}$. Otherwise, if $\hat{\eta}_t = \mathbf{0}$, it is a non-shifted attack.

This classification implies that an adversary will strategically allocate the poisoning budget $M$ to either increase the mean of $\eta_t$ or increase its variance.

When the adversary poisons the dataset $D_t$, the objective is to perturb the resulting model $w_t$ to maximize the cumulative excess risk across all tasks encountered up to time $t$. Conversely, the defender employs a regularization-based strategy, designing the regularizers $H_t$ to minimize this loss under adversarial influence. We formalize this strategic interaction as the following online minimax zero-sum game:

$$\min_{H_t} \max_{\hat{\eta}_t, \tilde{\eta}_t} \mathcal{R}_t(w_t) \ \text{ s.t. } \mathbb{E}[\|\hat{\eta}_t + \tilde{\eta}_t\|_F^2] \leq M, \ t \in [T], \tag{4}$$

where $M$ can be replaced by any of its upper bounds, without loss of generality. While heuristic defenses like adversarial training (Goodfellow et al., 2015) provide empirical utility, they generally lack formal performance guarantees. Conversely, we investigate the intrinsic robustness of regularization-based CL to derive rigorous limits on its efficacy against strategically crafted attacks, which can be integrated with these general defense mechanisms to further enhance robustness against poisoning attacks.

---

[1] We can also allow $K$ to scale sublinearly with $T$ when the noise $\tilde{\eta}_t$ is sub-Gaussian. Under the concentration properties of sub-Gaussian noise, we can further limit the adversary's poisoning in the task-to-task verification defense introduced in Section 4, ensuring convergence under sublinear attack tasks as stated in Theorem 4.4.

The objective in (4) is practically intractable, as neither the adversary nor the system has access to the ground-truth distributions $\mathcal{D}_\tau$ for tasks $\tau = 1, \ldots, t$. Moreover, even constructing an empirical counterpart of this objective is infeasible in CL: at each time $t$, both parties observe only the current dataset $D_t$, while the datasets from previous tasks are no longer accessible, making it impossible to form the cumulative empirical objective over tasks $1, \ldots, t$. To address this, we derive a surrogate for the inner adversarial optimization problem under the regularization-based framework in (2), as established in the following proposition.

**Proposition 2.4.** *Define $Z_t$ as the matrix formed by concatenating all samples in the dataset, and define training loss $\mathcal{L}(w, Z_t) = \sum_{z \in D_t} \ell(w, z)/n_t$. Let $w_t^*$ be a minimizer of $\mathcal{L}(w, Z_t)$, i.e., $w_t^* \in \arg\min_w \mathcal{L}(w, Z_t)$. Under a non-shifted attack $\tilde{\eta}_t$, the adversarial objective in (4) can be characterized via the following approximation:*

$$\max_{\tilde{\eta}_t} \frac{1}{2} \mathbb{E}\big[\|(\nabla_{ww}^2 \mathcal{L}_t + H_t)^+ \nabla_{wZ}^2 \mathcal{L}_t \operatorname{vec}(\tilde{\eta}_t)\|_{P_t}^2\big]$$

$$+ \mathbb{E}\left[O\big(\|\tilde{\eta}_t\|_F^2 + \|w_t - w_t^*\|_2^2\big) + O\left(\sum_{s=1}^t \|w_t - w_s^*\|_2^3\right)\right],$$

$$s.t. \ \mathbb{E}[\|\tilde{\eta}_t\|_F^2] \leq M, \ t \in [T], \tag{5}$$

*where we use the shorthand $\mathcal{L}_s = \mathcal{L}(w_s^*, Z_s)$, and $P_t := \sum_{s=1}^t \nabla_{ww}^2 \mathcal{L}_s$ represents the cumulative Hessian.*

The extension to the deterministic shift $\hat{\eta}_t$, along with its proof, is provided in Proposition A.1 of Appendix A. It is easy to see that an adversary's approximately optimal solutions to problem (5) allocate its covariance budget along the top right singular vector of $P_t^{1/2}(\nabla_{ww}^2 \mathcal{L}_t + H_t)^+ \nabla_{wZ}^2 \mathcal{L}_t$, strategically exploiting the sensitive features of the dataset to maximize the perturbation's impact on the model.

In (5), the remainder terms arise from terms beyond second order in the Taylor expansion of the empirical loss. When the loss is at most quadratic, all remainder terms vanish, and the approximation becomes exact for the empirical loss over all observed tasks. Moreover, in modern CL training, where models are typically highly overparameterized, the remainder terms in Proposition 2.4 are expected to be small. Consequently, the solution that maximizes the leading-order objective in (5) serves as a high-fidelity proxy for the true empirical optimum. This is supported by the literature on lazy training, which suggests that the task-specific optima $\{w_\tau^*\}$ and the iterates generated by (2) remain within a small neighborhood of the initialization (Jacot et al., 2018). In this regime, the Taylor residuals become negligible, and the approximation is asymptotically accurate.

To implement the attack in (5), an adversary can estimate the required parameters using the known model state $w_{t-1}$, together with efficient Hessian approximation techniques

such as K-FAC (Eschenhagen et al., 2023) or the Gauss-Newton method (Nocedal & Wright, 2006) to effectively calculate the required Hessian matrix.

# 3. Fundamental Limits for Defending Against Data Poisoning in CL

We next explore the fundamental boundaries of regularization-based robustness by evaluating the system's resilience to attacks in Definitions 2.1–2.3.

To theoretically quantify the robustness of regularization-based methods, we adopt the provable convergence of the excess risk (1) as the primary metric for defensibility in CL. We analyze a tractable setting with a uniform task distribution, i.e., $\mathcal{D}_1 = \cdots = \mathcal{D}_T$, to investigate defense against poisoning attacks. In Theorem 3.1, we demonstrate that convergence is not guaranteed under the proposed attack scenarios; by extension, this result precludes convergence in more general, non-uniform scenarios.

**Theorem 3.1.** *There exist problem instances where, for any regularization-based method in (2), no provable model convergence can be ensured in the uniform task distribution setting, under either of the following two attack scenarios:*

*(a) Frequent and shifted attacks;*

*(b) Frequent, unbounded, and non-shifted attacks.*

The proof of Theorem 3.1 is detailed in Appendix B, with the core intuition summarized as follows. For the first case, widespread distribution-shifting attacks induce an identifiability crisis, where malicious perturbations become statistically indistinguishable from legitimate task shifts. Consequently, designing the regularizer $H_t$ to ensure convergence guarantees comes into fundamental conflict with designing it to filter out adversarial attacks. For the second case, even with an optimal defense, excessive non-shifted poisoning can overwhelm the dataset's intrinsic features, preventing the excess risk from converging to zero.

Moving beyond these indefensible regimes, we focus our design and analysis on defense strategies against two possibly defensible classes of attacks: infrequent unbounded shifted attacks and frequent bounded non-shifted attacks. Notably, even in these scenarios, conventional regularization-based methods exhibit significant vulnerabilities, and a rigorous theoretical framework for defense remains elusive. Consequently, there is a critical need for advanced defense mechanisms that offer provable performance guarantees under attack.

# 4. Task-to-task Verification Defense Against Infrequent and Unbounded Attacks

In this section, we design and analyze the defense approach against the first possibly defensible case of infrequent yet unbounded poisoning attacks, where the attack can shift its data pattern using an unbounded poisoning budget $M$ in (5).

## 4.1. Design of Task-to-task Verification Defense

Defending against infrequent yet unbounded attacks presents two primary challenges. First, because the defender does not know in advance which tasks are attacked, a detection-based defense is necessary. In the regularization-based framework, detection acts as extreme regularization, with $H_t \to \infty$ freezing the update and rejecting the flagged task, thereby preventing unbounded shifted poisoning from entering the learning trajectory. This approach is grounded on the theoretical premise that the sequence of task-specific minimizers $\{w_t^*\}_{t=1}^T$ typically remains within a bounded proximity in the parameter space (Chizat et al., 2019; Guo et al., 2021; Bedi et al., 2019). Such stability implies that malicious interventions can be identified via the model update $\|w_t - w_{t-1}\|$. However, when a significant deviation occurs, the system faces an attribution dilemma: it cannot determine whether the large $\|w_t - w_{t-1}\|$ comes from a benign task correcting a model corrupted by consecutive past attacks or from a malicious task driving an otherwise benign update away from the clean learning trajectory.

To resolve the attribution dilemma, we design a Task-to-Task (T2T) verification defense. This mechanism leverages the dynamics of two consecutive parameter updates, $w_t - w_{t-1}$ and $w_{t-1} - w_{t-2}$ to compute a specialized detection score $d_t$ that eliminates the influence of all tasks prior to $t-1$, isolating the deviations introduced exclusively by tasks $t-1$ and $t$. This allows the system to detect the attack within the latest two tasks independently of the interference from earlier model states. Our dynamic design of $d_t$ is as follows:

$$d_t = \|D_t^1(w_t - w_{t-1}) - D_t^2(w_{t-1} - w_{t-2})\|, \quad (6)$$

where the projection matrices are given by $D_t^1 = (I - B_t)(A_t^+ - C_t A_t^\top)$ and $D_t^2 = (I - B_t)(B_t^+ + C_t B_t^\top)$. The coupling matrix $C_t$ is defined as:

$$C_t = (A_t^+ A_t - B_t^+ B_t)(A_t^\top A_t + B_t^\top B_t)^+, \quad (7)$$

with $A_t$ and $B_t$ constructed from the Hessian and regularizers:

$$A_t = \left(\nabla_{ww}^2 \mathcal{L}_t + H_t\right)^{-1} \nabla_{ww}^2 \mathcal{L}_t \left(\nabla_{ww}^2 \mathcal{L}_{t-1} + H_{t-1}\right)^{-1} H_{t-1}$$
$$B_t = \left(\nabla_{ww}^2 \mathcal{L}_{t-1} + H_{t-1}\right)^{-1} \nabla_{ww}^2 \mathcal{L}_{t-1}. \quad (8)$$

Geometrically, $d_t$ measures the projected updates after projecting $w_t - w_{t-1}$ and $w_{t-1} - w_{t-2}$ onto the largest common

subspace of the data features of tasks $t-1$ and $t$ and removing the influence of the historical state $w_{t-2}$. Consequently, the score reflects only the local dynamics of tasks $t-1$ and $t$, which we formally establish in the following lemma.

**Lemma 4.1.** *The dynamic detection score $d_t$ computed from* (6) *satisfies*

$$D_t^1(w_t - w_{t-1}) - D_t^2(w_{t-1} - w_{t-2}) =$$
$$E_t^1(w_t^* - w_{t-1}^*) + E_t^2 \operatorname{vec}(\eta_{t-1}) + E_t^3 \operatorname{vec}(\eta_t) + \mathcal{E}_t,$$

*where the remainder $\mathcal{E}_t$ satisfies*

$$\mathcal{E}_t = O\left(\|w_t - w_t^*\|^2\right) + O\left(\|w_{t-1} - w_{t-1}^*\|^2\right)$$
$$+ O\left(\|\eta_t\|^2 + \|\eta_{t-1}\|^2\right).$$

*Here, $\eta_s = \mathbf{0}$ if the adversary does not attack task $s$, and the explicit forms of the matrices $\{E_t^i\}_{i=1}^3$ are provided in Appendix C.1.*

The proof of Lemma 4.1 is provided in Appendix C.1. In general, the eigenvalues of $\{E_t^i\}_{i=1}^3$ and the model similarity metric $\|w_t^* - w_{t-1}^*\|$ remain stable under standard dataset normalization. Consequently, $d_t$ stays stable without attack ($\eta_{t-1}, \eta_t = \mathbf{0}$), whereas high-magnitude attacks of $\eta_{t-1}$ or $\eta_t$ trigger anomalous peaks on $d_t$. We operationalize this by flagging scores that significantly deviate from a rolling-window mean of length $h$. To resolve the attribution dilemma of identifying whether task $t-1$ or $t$ (or both) was poisoned, the system adopts a conservative recovery strategy by discarding both $w_t$ and $w_{t-1}$ upon detection. This T2T verification procedure is formalized in Alg. 1.

---

**Algorithm 1** Task-to-task Verification Defense
___

1: Initialize the parameter $w_0 = 0$ and $H_0 = 0$.
2: **for** $t = 1$ to $T$ **do**
3:     Update $w_t$ by (2).
4:     **if** the system has stored $w_{t-2}$ and $w_{t-1}$ **then**
5:         Compute dynamic detection score $d_t$ in (6).
6:         **if** $d_t \geq r \cdot \operatorname{mean}(d_{\max(1, t-h)}, \ldots, d_{t-1})$ **then**
7:             Set $w_t = w_{t-2}$ and $H_t = H_{t-2}$.
8:         **else**
9:             Keep $w_{t-1}, w_t, H_{t-1}, H_t$, and $\nabla_{ww}^2 \mathcal{L}_t$.
10:         **end if**
11:     **else**
12:         Keep $w_{t-1}, w_t, H_{t-1}, H_t$, and $\nabla_{ww}^2 \mathcal{L}_t$.
13:     **end if**
14:     Discard information that is not kept.
15: **end for**

---

As detailed in Alg. 1, the system maintains three consecutive models and their corresponding regularization terms, $\{w_s, H_s\}_{s=t-1}^t$ and $\nabla_{ww}^2 \mathcal{L}_t$. This introduces a constant storage overhead of $3p^2 + 2p$ to maintain data for the three most recent tasks, where $p$ denotes the parameter dimensionality. The memory associated with the Hessians can be

further mitigated by employing approximation, such as diagonal estimation, which reduces the overhead from $O(p^2)$ to $O(p)$.

### 4.2. Theoretical Analysis of Task-to-task Verification Defense

We next establish rigorous performance guarantees for our T2T verification defense in Alg. 1 under the continual linear regression setting. This typical setting has gained significant attention and is widely studied in recent theoretical CL literature to facilitate the derivation of a closed-form theoretical framework (Evron et al., 2022; Lin et al., 2023; Li et al., 2023; Swartworth et al., 2023; Zhao et al., 2024). The continual linear regression problem offers valuable insights into general non-linear CL models because, under the Neural Tangent Kernel (NTK) regime, general non-linear models behave like linear models (Jacot et al., 2018; Lee et al., 2019). Furthermore, many modern CL methods utilize pre-trained models and optimize only an additional linear layer (Peng et al., 2025), which simplifies the setting to an equivalent linear problem. The detailed model setup is provided in Assumption 4.2.

**Assumption 4.2.** For each task $t$, the clean observations follow a multi-output linear model

$$Y_t = X_t w^* + \varepsilon_t,$$

where $X_t = [x_t^1, \ldots, x_t^{n_t}]^\top \in \mathbb{R}^{n_t \times p_1}$ is the feature matrix, $Y_t = [y_t^1, \ldots, y_t^{n_t}]^\top \in \mathbb{R}^{n_t \times p_2}$ is the response matrix, and $w^* \in \mathbb{R}^{p_1 \times p_2}$ is the shared ground-truth parameter with $\|w^*\|_F^2 \leq W$. The noise matrix $\varepsilon_t = [\varepsilon_t^1, \ldots, \varepsilon_t^{n_t}]^\top \in \mathbb{R}^{n_t \times p_2}$ has independent rows satisfying $\mathbb{E}[\varepsilon_t^i \mid x_t^i] = 0$, $\mathbb{E}[\varepsilon_t^i (\varepsilon_t^i)^\top \mid x_t^i] = \sigma^2 I_{p_2}$. The adversary conducts label poisoning only, so when task $t$ is attacked the training responses become $X_t w^* + \varepsilon_t + \eta_t$, while the feature matrix $X_t$ remains unchanged. We use the squared loss $\ell(w, (x, y)) = \|x^\top w - y\|_2^2/2$. The parameter-space excess risk is defined as

$$\mathcal{R}(w_t) := \mathbb{E}[\|w_t - w^*\|_F^2].$$

Under the model in Assumption 4.2, we establish a theoretical threshold $\theta_t$ for the detection score $d_t$ defined in (6) to ensure a more theoretically grounded detection mechanism. This threshold ensures two critical properties: first, a benign task will not be erroneously filtered out with high probability; second, any attack that remains below $\theta_t$ is guaranteed not to compromise the convergence of the CL process. This theoretical foundation allows us to replace the heuristic detection condition in Alg. 1 with the formal condition $d_t \geq \theta_t$, thereby providing a more reliable detection mechanism and facilitating a rigorous theoretical analysis of the method's performance. We first derive a high-probability upper bound for the detection score $d_t$ in the absence of adversarial intervention to serve as the detection threshold.

**Lemma 4.3.** *Under Assumption 4.2, when tasks $t-1$ and $t$ are benign, the detection score $d_t$ satisfies $d_t \leq \theta_t$ for all $t = 2, \ldots, T$ with probability at least $1 - \epsilon$, where:*

$$\theta_t = \sqrt{\frac{\sigma^2 T}{\epsilon} \left[ \text{tr}\left((E_t^2)^\top E_t^2\right) + \text{tr}\left((E_t^3)^\top E_t^3\right) \right]}.$$

Lemma 4.3 establishes a distribution-free guarantee that remains robust under arbitrary distributions of the inherent noise $\varepsilon_t$. When the noise exhibits stronger concentration, such as sub-Gaussianity, Corollary C.2 in Appendix C.2 demonstrates that the detection threshold can be significantly tightened. Proofs for both results are deferred to Appendix C.2.

We now formally characterize the generalization performance of Alg. 1. The following theorem establishes that under the proposed T2T verification defense, the final model error will converge to zero as the number of tasks $T$ increases, in the presence of $K$ adversarial tasks.

**Theorem 4.4.** *Suppose Alg. 1 replaces its heuristic detection condition with $d_t > \theta_t$, where $\theta_t$ is defined in Lemma 4.3, and uses the regularization-based update in (2) with the following regularizer $H_t$:*

$$H_t = \frac{1}{n_t} \left( \frac{\sigma^2}{W} I + \sum_{s=1}^{t-1} \mathbf{X}_s^\top \mathbf{X}_s \right), \quad \mathbf{X}_s = I_{p_2} \otimes X_s. \quad (9)$$

*Then, the final excess risk $\mathcal{R}(w_T) = O\left(\frac{K^2}{\epsilon T}\right)$ if the datasets for all tasks are informative [2]. Let $\tilde{\mathbf{X}}_t = \left( \frac{\sigma^2}{W} I + \sum_{s=1}^{t-1} \mathbf{X}_s^\top \mathbf{X}_s \right)$, the explicit upper bound is given by:*

$$\mathcal{R}(w_T) \leq K \sum_{i=1}^{K} \frac{\sigma^2 T}{\epsilon} \left\| \tilde{\mathbf{X}}_{T+1}^{-1} (\mathbf{X}_{\tau_i}^\top \mathbf{X}_{\tau_i}) \right\|_2^2 tr((\mathbf{X}_{\tau_i}^\top \mathbf{X}_{\tau_i})^+)$$
$$+ \sigma^2 tr(\tilde{\mathbf{X}}_{T+1}^{-1}), \quad (10)$$

*where $\tau_i$ for $i \in [K]$ is the time index of poisoned tasks.*

The proof of Theorem 4.4 is given in Appendix C.3. The second term in (10) represents the excess risk for benign tasks, while the first term quantifies the error introduced by the shifted attack, which scales as $O(K^2/T)$. Consequently, we provide a defense mechanism with provable performance guarantees.

## 5. Robust Feature Defense Against Frequent, Bounded, and Non-shifted Attacks

In this section, we investigate defense mechanisms against the second category of possibly defensible attacks: frequent,

---

[2]We define a dataset as informative if, for every dimension, a linear proportion of the total tasks possesses a non-zero eigenvalue. Without this assumption, the excess risk fails to converge even in the absence of an attack.

bounded, and non-shifted poisoning. This attack type is relatively more tractable than unbounded or shifted attacks, because the perturbations are bounded and do not induce distributional shifts. Hence, an algorithm that converges under stochastic noise should, in principle, still be able to learn under such poisoning.

However, unlike inherent stochastic noise, which is typically independent and non-adaptive, an adversary strategically crafts $\tilde{\eta}_t$ to maximize its impact on the learning trajectory, as in (5). This shift from random noise to targeted intervention necessitates a framework that explicitly quantifies how adversarial perturbations slow convergence, moving beyond passive noise robustness to account for strategic attacks.

Under a regime of pervasive poisoning, the task-to-task verification mechanism in Alg. 1 becomes untenable. In scenarios where nearly every task is compromised, a rejection-based strategy would lead to the discarding of the entire task sequence, resulting in a total loss of system utility. This necessitates an intrinsic redesign of the regularization framework in (2) by optimizing the regularization term $H_t$ in the attack-defense model in (11). Based on Proposition 2.4 and further incorporating the system's objective of choosing $H_t$ to minimize the excess risk in (1), we derive the following tractable surrogate optimization problem:

$$\min_{H_t} \max_{\tilde{\eta}_t} \frac{1}{2}\mathbb{E}\big[\|(\nabla_{ww}^2\mathcal{L}_t + H_t)^+ \nabla_{wZ}^2 \mathcal{L}_t \operatorname{vec}(\tilde{\eta}_t)\|_{P_t}^2\big] +$$

$$\frac{1}{2}\sum_{s=1}^{t}\mathbb{E}\left[\|\nabla_{ww}^2\mathcal{L}_t\cdot(w_t^*-w_s^*)+H_t(w_{t-1}-w_s^*)\|_{\tilde{P}_s}^2\right]$$

$$+ \mathbb{E}\left[O(\|\tilde{\eta}_t\|_F^2 + \|w_t - w_t^*\|_2^2)\right]$$

$$+ \mathbb{E}\left[O\left(\sum_{s=1}^{t}\|w_t - w_s^*\|_2^3\right)\right],$$

$$\text{s.t. } \mathbb{E}[\|\tilde{\eta}_t\|_F^2] \le M, \tag{11}$$

where $\tilde{P}_s := (\nabla_{ww}^2\mathcal{L}_t + H_t)^+ \nabla_{ww}^2\mathcal{L}_s(\nabla_{ww}^2\mathcal{L}_t + H_t)^+$.

The detailed derivation of this surrogate problem is provided in Appendix A. This defense model suggests how the system must proactively reshape the regularization term $H_t$ to account for potential perturbations within the dataset $D_t$ under adversarial awareness. Given that the adversary's optimal strategy in the inner maximization problem is to concentrate the poisoning budget along the principal eigenvector to maximize the degradation of the generalization risk, an effective defense should prioritize suppressing the maximum singular value of the sensitivity matrix $P_t^{1/2}(\nabla_{ww}^2\mathcal{L}_t + H_t)^+\nabla_{wZ}^2\mathcal{L}_t$. In this regime, the system must optimize $H_t$ by balancing three competing error sources: task dissimilarity $(w_t^* - w_s^*)$, historical model drift $w_{t-1} - w_s^*$, and adversarial noise $\eta_t$.

Directly solving the approximate problem in (11) is non-trivial, as it requires the estimation of multiple high-dimensional objective terms. To derive actionable insights

and provide rigorous performance guarantees, we further analyze this defense framework under the linear CL.

### 5.1. Theoretical Analysis of Our Robust Feature Defense

Under Assumption 4.2, we vectorize the variables and, with a slight abuse of notation, write $w_t \leftarrow \operatorname{vec}(w_t)$, $\tilde{\eta}_t \leftarrow \operatorname{vec}(\tilde{\eta}_t)$, $X_t \leftarrow I_{p_2} \otimes X_t$. Then, the system's optimization in (11) reduces to the following under Assumption 4.2 for minimizing $\mathcal{R}(w_t) = \mathbb{E}[\|w_t - w_*\|_2^2]$:

$$\min_{H_t} \max_{\operatorname{Cov}(\tilde{\eta}_t)\succeq 0} \left\|S_t^{-1}H_t\mathbb{E}[w_{t-1} - w_*]\right\|_2^2$$

$$+ \operatorname{tr}\left(S_t^{-1}\frac{X_t^\top}{n_t}(\operatorname{Cov}(\tilde{\eta}_t) + \sigma^2 I)\frac{X_t}{n_t}S_t^{-1}\right)$$

$$+ \operatorname{tr}\left(S_t^{-1}H_t\operatorname{Cov}(w_{t-1} - w_*)H_t^\top S_t^{-1}\right)$$

$$\text{s.t. } \operatorname{tr}(\operatorname{Cov}(\tilde{\eta}_t)) \le M, \tag{12}$$

where $S_t = \frac{X_t^\top X_t}{n_t} + H_t$ is symmetric and invertible by choosing $H_t$. The corresponding recursions for the mean and covariance under $H_t$ and $\tilde{\eta}_t$ are

$$\mathbb{E}[w_t - w_*] = S_t^{-1}H_t\mathbb{E}[w_{t-1} - w_*],$$

$$\operatorname{Cov}(w_t - w_*) = S_t^{-1}\frac{X_t^\top}{n_t}(\operatorname{Cov}(\tilde{\eta}_t) + \sigma^2 I)\frac{X_t}{n_t}S_t^{-1}$$

$$+ S_t^{-1}H_t\operatorname{Cov}(w_{t-1} - w_*)H_t^\top S_t^{-1} \tag{13}$$

for $t = 1, \dots, T$. If $w_0 = 0$, then $\|\mathbb{E}[w_0 - w_*]\|_2^2 = \|w_*\|_2^2 \le W$ and $\operatorname{Cov}(w_0 - w_*) = \mathbf{0}$.

The derivations of (12) and (13) are detailed in Appendix D.1. Under this recursion, the defense can be implemented recursively: at each time $t$, the system solves (12) to determine the defender's regularizer $H_t$ together with the worst-case adversarial strategy, and then substitutes both strategies into the recursions to update the estimates of $\mathbb{E}[w_t - w_*]$ and $\operatorname{Cov}(w_t - w_*)$. As a result, even though the system does not know which tasks are attacked or which admissible perturbations are used, it obtains a uniform excess-risk upper bound by optimizing against the worst-case scenario in which every task is treated as potentially attacked and the adversary maximizes the excess-risk bound at each time.

To derive a closed-form solution for $H_t$ and provide further insight, we impose the following standard assumption in addition to Assumption 4.2, consistent with the existing theoretical literature on CL (Li et al., 2023; Wu et al., 2022; Zhao et al., 2024).

**Assumption 5.1.** The matrices $\{\mathbf{X}_s^\top \mathbf{X}_s\}_{s=1}^{T}$ pairwise commute.

Under Assumption 5.1, the task Hessians are simultaneously diagonalizable. Thus, there exists an orthogonal matrix

$U = [u_1, \ldots, u_p]$ such that

$$\frac{X_t^\top X_t}{n_t} = U\Gamma_t U^\top, \ \Gamma_t = \mathrm{diag}(\gamma_1^t, \ldots, \gamma_p^t).$$

It is therefore sufficient to restrict the regularizer to the same eigenbasis, $H_t = U\Lambda_t U^\top$, $\Lambda_t = \mathrm{diag}(\lambda_1^t, \ldots, \lambda_p^t)$, so that the defense design reduces to choosing the scalar regularization strengths $\{\lambda_j^t\}_{j=1}^p$.

This diagonalization also decomposes the parameter-space excess risk into independent eigendirections: $\mathcal{R}(w_t) = \sum_{j=1}^p \mathcal{R}_j^t$, where $\mathcal{R}_j^t = \mathbb{E}[(u_j^\top(w_t - w_*))^2]$ denotes the contribution along eigenvector $u_j$. Under the worst-case setting where every task may be poisoned, the minimax problem in (12) reduces to a scalar per-direction recursion for $\mathcal{R}_j^t$. The full recursion and derivation are deferred to Appendix D.1.

We derive the optimal eigenvalue design $\Lambda_t = \mathrm{diag}(\lambda_1^t, \ldots, \lambda_p^t)$, equivalently the Nash equilibrium of the minimax recursion in (12), in the following proposition.

**Proposition 5.2.** *Let $\tilde{\gamma}_j^t := \gamma_j^t n_t$ for all $t \in [T]$, and define $\mathcal{I}_t^+ := \{j \in [p] : \tilde{\gamma}_j^t > 0, \ \mathcal{R}_j^{t-1} > 0\}$. For $j \in \mathcal{I}_t^+$, define $b_j^t := \frac{\mathcal{R}_j^{t-1}\tilde{\gamma}_j^t + \sigma^2}{\mathcal{R}_j^{t-1}\sqrt{\tilde{\gamma}_j^t}}$. Reorder the indices in $\mathcal{I}_t^+$ such that $b_1^t \le \cdots \le b_{|\mathcal{I}_t^+|}^t$, breaking ties arbitrarily. For $j \le |\mathcal{I}_t^+|$, define $A_j^t := \frac{M + j\sigma^2 + \sum_{k=1}^j \mathcal{R}_k^{t-1}\tilde{\gamma}_k^t}{\sum_{k=1}^j \mathcal{R}_k^{t-1}\sqrt{\tilde{\gamma}_k^t}}$. If $\mathcal{I}_t^+ = \emptyset$, set $j_t = 0$. Otherwise, define $j_t$ as the maximum value in the following set:*

$$J_t = \{j \in \{1, \ldots, |\mathcal{I}_t^+|\} : A_j^t > b_j^t\}. \tag{14}$$

*If the set above is empty, set $j_t = 0$. The system's best defense strategy of $\lambda_j^t$ is*

$$\lambda_j^t = \begin{cases} \dfrac{\sigma^2}{n_t \mathcal{R}_j^{t-1}}, & j \in \mathcal{I}_t^+, \ j > j_t, \\[2ex] A_{j_t}^t \sqrt{\dfrac{\gamma_j^t}{n_t}} - \gamma_j^t, & j \in \mathcal{I}_t^+, \ j \le j_t, \\[2ex] +\infty, & \gamma_j^t > 0, \ \mathcal{R}_j^{t-1} = 0, \\[1ex] \text{any positive value}, & \gamma_j^t = 0. \end{cases} \tag{15}$$

*Here the first two cases are stated after the above reordering of the indices in $\mathcal{I}_t^+$.*

The proof of Proposition 5.2 is given in Appendix D.2. Under this strategy, the set of dimensions $\{1, \ldots, j_t\}$ in the Hessian space represents the "Protected Set," where the adversary allocates most of their attack budget; consequently, the system specifically tailors the regularization parameters $\{\lambda_j^t\}$ to bolster the defenses of these vulnerable dimensions.

Zhao et al. (2024) proves that the regularization term $H_t$ defined in (9), equivalent to the EWC algorithm (Kirkpatrick

et al., 2017b), minimizes the upper bound of $\mathcal{R}(w_t)$ for each task without attacks under Assumption 4.2. We adopt this formulation as a benchmark to illustrate how our method accelerates the convergence of $\mathcal{R}(w_T)$ by reducing the adversarial sensitivity of the data features. While the per-dimension iteration of $\mathcal{R}_j^t$ in (43) is complex and lacks analytical expression, we establish an asymptotic upper bound for the total excess risk $\sum_{j=1}^p \mathcal{R}_j^T$ under a special case.

**Theorem 5.3.** *Consider the special stationary problem instance specified in Theorem D.1, in which the protected dimensions $\{1, \ldots, j_T\}$ selected by Proposition 5.2 remain fixed over time. Then our robust feature defense in (15) ensures fast convergence with the following excess risk for some constants $\{f_j\}_{j=1}^{j_T}$ as $T \to \infty$:*

$$\mathcal{R}(w_T) \lesssim \frac{M + j_T \sigma^2}{T\left(\sum_{j=1}^{j_T} f_j \sqrt{\tilde{\gamma}_j^T}\right)^2} + \sum_{j=j_T+1}^p \frac{\sigma^2}{\sigma^2/W + \sum_{\tau=1}^T \tilde{\gamma}_j^\tau}, \tag{16}$$

*which improves over the following excess risk of the latest regularization-based method in (Zhao et al., 2024) using $H_t$ in (9):*

$$\mathcal{R}(w_T) \lesssim \max_j \frac{M}{T\tilde{\gamma}_j^T} + \sum_{j=1}^p \frac{\sigma^2}{\sigma^2/W + \sum_{\tau=1}^T \tilde{\gamma}_j^\tau}. \tag{17}$$

The complete version of Theorem 5.3 and the proof are given in Appendix D.3. In (17), the attack-induced term is determined by a worst-direction maximum, $\max_j M/(T\tilde{\gamma}_j^T)$. Thus the convergence rate can be dominated by a single highly sensitive eigendirection with small $\tilde{\gamma}_j^T$. Our robust feature defense effectively spreads this worst-case sensitivity over the protected set. This redistribution prevents the attack-induced loss from being dictated by a single small-eigenvalue direction and yields a more stable convergence rate under imbalanced spectra.

# 6. Experimental Validation

In this section, we evaluate our attack-defense framework across linear and non-linear models to reflect real-world scenarios. Following a widely adopted protocol (Peng et al., 2025), we utilize CIFAR-100 with a pretrained Vision Transformer (ViT) (Dosovitskiy et al., 2021), where only an additional linear layer is trained. To assess non-linear dynamics, we train a Convolutional Neural Network from scratch on CIFAR-10 due to computational constraints. Both datasets are partitioned into 100 tasks for CL. We benchmark our proposed defense against two established baselines: regularization-based EWC (Kirkpatrick et al., 2017b) and memorization-based iCaRL (Rebuffi et al., 2017). Implementation details are provided in Appendix E.

We first simulate the infrequent shifted attacks in Section 4 by randomly poisoning 10 out of 100 tasks. For the CIFAR-100 task, the lower subfigure of Fig. 2a shows that the test accuracy under our Alg. 1 remains comparable to the no-attack baseline after discarding the detected tasks. Conversely, pure EWC and iCaRL, lacking carefully designed defense mechanisms, converge slower under shifted attacks.

Moreover, the upper subfigure of Fig. 2a shows that the detection scores exhibit prominent peaks upon attack, all of which are successfully identified by the criterion established in Alg. 1, while the accuracy in the lower subfigure of Fig. 2a remains deceptively stable under attack and lacks a signature of malicious intervention. This shows that although data poisoning is often imperceptible through macroscopic metrics like accuracy, our Alg. 1 effectively exposes these threats by providing a clear microscopic indicator.

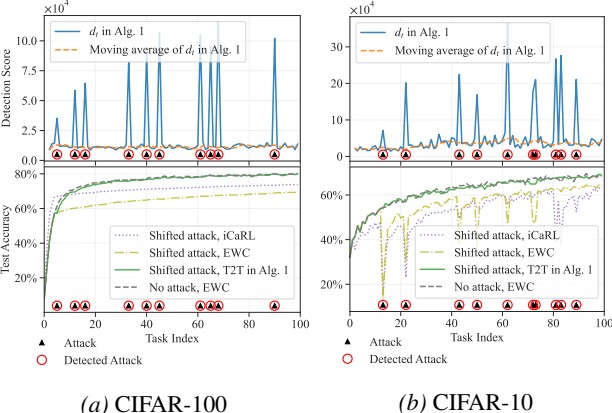

*(a)* CIFAR-100      *(b)* CIFAR-10

*Figure 2.* Performance evaluation under infrequent shifted attacks on CIFAR-100 (left) and CIFAR-10 (right). The upper panels show the evolution of the detection score $d_t$ in (6) across tasks, and the lower panels show the test accuracies of the attack-free baseline, EWC, iCaRL, and EWC integrated with our T2T defense. Black triangles indicate detected attacks, while red circles indicate the ground-truth attacked tasks.

Fig. 2b further illustrates the results on the CIFAR-10 dataset with a non-linear model. In this non-linear setting, all adversarial interventions are successfully identified by our mechanism, validating that our Alg. 1 remains effective across both linear and complex non-linear models. Note that the test accuracy for CIFAR-100 remains more stable under attack than that of CIFAR-10. This is because the frozen parameters in the pretrained model already possess feature-extraction capabilities, whereas the CIFAR-10 model trained from scratch lacks this prior knowledge.

Figs. 2a and 2b show that iCaRL outperforms EWC under shifted attacks. This resilience occurs because iCaRL trains on a mix of the current task and clean exemplars. Since the attack only targets the current task, these benign exemplars prevent the model from fully drifting in the ad-

versarial direction. This highlights a potential advantage for memorization-based methods against poisoning.

We next evaluate CL under the frequent and bounded non-shifted attacks described in Section 5. On synthetic data, we simulate the Monte Carlo attack multiple times and report the average performance of our robust feature defense, which recursively solves (12) without Assumption 5.1. Fig. 4 in Appendix E confirms the faster excess-risk convergence predicted by Theorem 5.3, compared with the benchmark of $H_t$ in (9) from (Zhao et al., 2024).

We then evaluate the CIFAR-100 setting and assess the efficacy of our proposed defense mechanism derived from (11) in comparison to the standard EWC and iCaRL algorithms. The results in Fig. 3 demonstrate that our defense effectively accelerates convergence and mitigates accuracy degradation under the strategic attack, confirming its ability to stabilize the learning trajectory against pervasive adversarial noise.

In contrast to its success against infrequent attacks in Figs. 2a and 2b, iCaRL performs significantly worse than regularization-based methods under frequent attacks (Fig. 3). Without a regularization term, iCaRL is more sensitive to the adversarial objective in (5). This allows the adversary to disrupt training more effectively by targeting the model's most sensitive features with strategic noise.

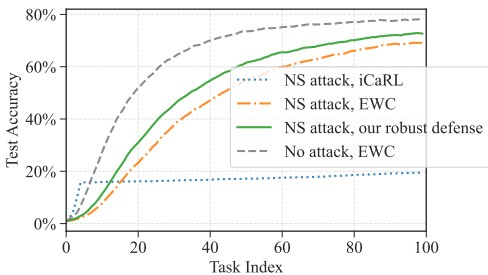

*Figure 3.* Test accuracy during CL on CIFAR-100 under four scenarios: (i) the strategic attack targeting iCaRL; (ii) the strategic attack targeting EWC; (iii) the strategic attack countered by robust feature defense in (11); and (iv) attack-free.

# 7. Conclusion

In this work, we establish a first theoretical foundation for data poisoning attacks and defenses in regularization-based CL. By modeling the adversary-defender interaction as an online zero-sum game, we propose a T2T defense against infrequent attacks and a robust feature defense against frequent non-shifted attacks. In future work, we will further investigate how to extend our theory to large-scale neural networks for more practical implementations, and broaden the study of data poisoning attacks to other CL frameworks, such as replay-based CL.

## Acknowledgments

This work was supported in part by the Guangdong Provincial Key Lab of Integrated Communication, Sensing and Computation for Ubiquitous Internet of Things (No. 2023B1212010007).

## Impact Statement

This paper presents work whose goal is to advance the field of machine learning. There are many potential societal consequences of our work, none of which we feel must be specifically highlighted here.

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

## A. Missing details from Section 2 and (11)

**Proposition A.1.** *Define $Z_t$ as the matrix formed by concatenating all samples in the dataset, and define $\mathcal{L}(w, Z_t) = \sum_{z \in D_t} \ell(w, z)/n_t$. Let $w_t^*$ be a minimizer of the training loss for task $t$, such that $w_t^* \in \arg\min_w \mathcal{L}(w, Z_t)$. At time $t$, the adversarial objective defined in (4) can be approximated as:*

$$
\max_{\hat{\eta}_t, \tilde{\eta}_t} \mathbb{E}\left[\frac{1}{2}\|(\nabla_{ww}^2 \mathcal{L}(w_t^*, Z_t) + H_t)^+ \nabla_{wZ}^2 \mathcal{L}(w_t^*, Z_t) \operatorname{vec}(\tilde{\eta}_t + \hat{\eta}_t)\|_{P_t}^2\right]
$$

$$
- \mathbb{E}\left[\operatorname{vec}(\hat{\eta}_t)^\top \nabla_{wZ}^2 \mathcal{L}(w_t^*, Z_t)^\top (\nabla_{ww}^2 \mathcal{L}(w_t^*, Z_t) + H_t)^+\right.
$$

$$
\left. \cdot \left(\sum_{\tau=1}^t \nabla_{ww}^2 \mathcal{L}(w_\tau^*, Z_\tau)(\nabla_{ww}^2 \mathcal{L}(w_t^*, Z_t) + H_t)^+ \left(\nabla_{ww}^2 \mathcal{L}(w_t^*, Z_t)(w_t^* - w_\tau^*) + H_t(w_{t-1} - w_\tau^*)\right)\right)\right]
$$

$$
+ \mathbb{E}\left[O\left(\|\tilde{\eta}_t + \hat{\eta}_t\|_F^2 + \|w_t - w_t^*\|_2^2\right) + O\left(\sum_{\tau=1}^t \|w_t - w_\tau^*\|_2^3\right)\right]
$$

$$
\text{s.t. } \mathbb{E}[\|\tilde{\eta}_t + \hat{\eta}_t\|_F^2] \le M, \tag{18}
$$

*where $P_t = \sum_{s=1}^t \nabla_{ww}^2 \mathcal{L}(w_s^*, Z_s)$ represents the cumulative Hessian across all observed tasks, and $H_t$ is the regularization term from (2).*

*Proof of Proposition A.1.* For notational simplicity, denote

$$
Q_s = \nabla_{ww}^2 \mathcal{L}(w_s^*, Z_s), \quad G_t = \nabla_{wZ}^2 \mathcal{L}(w_t^*, Z_t), \quad S_t = Q_t + H_t,
$$

and let $\eta_t = \hat{\eta}_t + \tilde{\eta}_t$. Throughout the proof, $A^+$ denotes the Moore–Penrose pseudoinverse; when $S_t$ is singular, the local linearized update is understood as the minimum-norm solution selected by $S_t^+$.

We first replace the excess risk $\mathcal{R}_t(w_t)$ by its empirical counterpart. Hence the adversary's inner objective can be approximated by

$$
\max_{\eta_t} \sum_{\tau=1}^t \mathcal{L}(w_t, Z_\tau) \quad \text{s.t. } \mathbb{E}[\|\eta_t\|_F^2] \le M.
$$

By a second-order Taylor expansion around $w_\tau^*$, we have

$$
\sum_{\tau=1}^t \mathcal{L}(w_t, Z_\tau)
$$

$$
= \sum_{\tau=1}^t \left(\mathcal{L}(w_\tau^*, Z_\tau) + \nabla_w \mathcal{L}(w_\tau^*, Z_\tau)^\top (w_t - w_\tau^*) + \frac{1}{2}(w_t - w_\tau^*)^\top Q_\tau (w_t - w_\tau^*)\right) + O\left(\sum_{\tau=1}^t \|w_t - w_\tau^*\|_2^3\right)
$$

$$
= \sum_{\tau=1}^t \mathcal{L}(w_\tau^*, Z_\tau) + \frac{1}{2}\sum_{\tau=1}^t (w_t - w_\tau^*)^\top Q_\tau (w_t - w_\tau^*) + O\left(\sum_{\tau=1}^t \|w_t - w_\tau^*\|_2^3\right), \tag{19}
$$

where the last equality follows from the first-order optimality condition of $w_\tau^*$.

When task $t$ is poisoned by $\eta_t$, the update rule in (2) gives

$$
\nabla_w \mathcal{L}(w_t, Z_t + \eta_t) + H_t(w_t - w_{t-1}) = 0.
$$

Expanding the first term around $(w_t^*, Z_t)$, and using $\nabla_w \mathcal{L}(w_t^*, Z_t) = 0$, yields

$$
Q_t(w_t - w_t^*) + G_t \operatorname{vec}(\eta_t) + H_t(w_t - w_{t-1}) + O(\|w_t - w_t^*\|_2^2 + \|\eta_t\|_F^2) = 0. \tag{20}
$$

Therefore,

$$
w_t - w_\tau^* = S_t^+\left(Q_t(w_t^* - w_\tau^*) + H_t(w_{t-1} - w_\tau^*) - G_t \operatorname{vec}(\eta_t)\right) + O(\|w_t - w_t^*\|_2^2 + \|\eta_t\|_F^2). \tag{21}
$$

Substituting (21) into (19) and discarding terms independent of $\eta_t$, we obtain

$$\frac{1}{2}\left\|S_t^+ G_t \operatorname{vec}(\eta_t)\right\|_{P_t}^2 - \operatorname{vec}(\eta_t)^\top G_t^\top S_t^+ \left(\sum_{\tau=1}^t Q_\tau S_t^+ \left(Q_t(w_t^* - w_\tau^*) + H_t(w_{t-1} - w_\tau^*)\right)\right)$$

$$+ O(\|\eta_t\|_F^2 + \|w_t - w_t^*\|_2^2) + O\left(\sum_{\tau=1}^t \|w_t - w_\tau^*\|_2^3\right), \tag{22}$$

where $P_t = \sum_{\tau=1}^t Q_\tau$.

Finally, decompose $\eta_t = \hat{\eta}_t + \tilde{\eta}_t$. Since the non-shifted perturbation satisfies

$$\mathbb{E}\left[\operatorname{vec}(\tilde{\eta}_t) \mid Z_{1:t}, w_{t-1}\right] = 0,$$

the linear term involving $\tilde{\eta}_t$ vanishes after taking expectation. Keeping the attack-dependent terms gives the desired approximation in (18). $\square$

*Derivation of defense optimization problem* (11): By substituting the update rule (21) with only $\tilde{\eta}_t$, we have

$$\mathbb{E}\left[\sum_{\tau=1}^t \mathcal{L}(w_t, Z_\tau)\right]$$

$$= \frac{1}{2}\sum_{\tau=1}^t \mathbb{E}\left[\left\|S_t^+ \left(Q_t(w_t^* - w_\tau^*) + H_t(w_{t-1} - w_\tau^*) - G_t \operatorname{vec}(\tilde{\eta}_t)\right) + O(\|w_t - w_t^*\|_2^2 + \|\tilde{\eta}_t\|_F^2)\right\|_{Q_\tau}^2\right]$$

$$+ \mathbb{E}\left[O\left(\sum_{\tau=1}^t \|w_t - w_\tau^*\|_2^3\right) + \sum_{\tau=1}^t \mathcal{L}(w_\tau^*, Z_\tau)\right]$$

$$= \mathbb{E}\left[\frac{1}{2}\sum_{\tau=1}^t \|Q_t(w_t^* - w_\tau^*) + H_t(w_{t-1} - w_\tau^*)\|_{(S_t^+)^\top Q_\tau S_t^+}^2 + \frac{1}{2}\sum_{\tau=1}^t \left\|S_t^+ G_t \operatorname{vec}(\tilde{\eta}_t)\right\|_{(S_t^+)^\top Q_\tau S_t^+}^2\right]$$

$$+ \mathbb{E}\left[O(\|w_t - w_t^*\|_2^2 + \|\tilde{\eta}_t\|_F^2) + O\left(\sum_{\tau=1}^t \|w_t - w_\tau^*\|_2^3\right) + \sum_{\tau=1}^t \mathcal{L}(w_\tau^*, Z_\tau)\right].$$

After taking expectation, the cross term vanishes because $\mathbb{E}[\operatorname{vec}(\tilde{\eta}_t) \mid Z_{1:t}, w_{t-1}] = 0$. Discarding the last term, which is independent of $H_t$ and $\eta_t$, gives (11).

## B. Proof of Theorem 3.1

*Proof of Theorem 3.1.* It suffices to construct a one-dimensional continual linear regression instance. Let $p_1 = p_2 = n_t = 1$, $x_t = 1$, and

$$y_t = w^* + \varepsilon_t,$$

where $\mathbb{E}[\varepsilon_t] = 0$ and $\mathbb{E}[\varepsilon_t^2] = \sigma^2$. Choose the squared loss $\ell(w, z_t) = (w - y_t)^2$. In this scalar case, the regularization-based update in (2) takes the form

$$w_t = \frac{2}{2 + H_t}(y_t + \eta_t) + \frac{H_t}{2 + H_t} w_{t-1}.$$

Absorbing the constant factor into the regularization parameter, we equivalently write

$$w_t = \frac{1}{1 + H_t}(y_t + \eta_t) + \frac{H_t}{1 + H_t} w_{t-1}.$$

Define

$$c_t := \frac{H_t}{1 + H_t} \in [0, 1].$$

Then

$$w_t - w^* = c_t(w_{t-1} - w^*) + (1 - c_t)(\varepsilon_t + \eta_t). \tag{23}$$

We first consider frequent shifted attacks. Let the adversary poison every task, which is a frequent attack, and set $\eta_t = \Delta$ for a fixed nonzero constant $\Delta$. This attack has bounded per-task budget $M = \Delta^2$. Taking expectation in (23) and denoting $m_t := \mathbb{E}[w_t - w^*]$, gives

$$m_t = c_t m_{t-1} + (1 - c_t)\Delta \Rightarrow$$
$$m_t - \Delta = c_t(m_{t-1} - \Delta) \Rightarrow$$
$$m_T = \Delta + \prod_{t=1}^{T} c_t(w_0 - w^* - \Delta).$$

If $\prod_{t=1}^{T} c_t \not\to 0$, then even in the attack-free noiseless case the initial bias $w_0 - w^*$ cannot be eliminated, so convergence cannot be guaranteed. If $\prod_{t=1}^{T} c_t \to 0$, then under the shifted attack

$$m_T \to \Delta,$$

and therefore, by Jensen's inequality,
$$\mathbb{E}[(w_T - w^*)^2] \geq m_T^2 \to \Delta^2 > 0.$$

Thus no regularization-based method can guarantee convergence under frequent shifted attacks.

We next consider frequent, unbounded, non-shifted attacks. Again let the adversary poison every task, but now choose independent zero-mean perturbations $\tilde{\eta}_t$ with

$$\mathbb{E}[\tilde{\eta}_t] = 0, \qquad \mathbb{E}[\tilde{\eta}_t^2] = M_T,$$

where $M_T = \Omega(T)$. This is a non-shifted attack with unbounded per-task budget. Unrolling (23) gives

$$w_T - w^* = \left( \prod_{r=1}^{T} c_r \right) (w_0 - w^*) + \sum_{s=1}^{T} (1 - c_s) \left( \prod_{r=s+1}^{T} c_r \right) (\varepsilon_s + \tilde{\eta}_s).$$

Let

$$\beta_{s,T} := (1 - c_s) \prod_{r=s+1}^{T} c_r.$$

The weights satisfy the telescoping identity

$$\sum_{s=1}^{T} \beta_{s,T} = 1 - \prod_{r=1}^{T} c_r.$$

If $\prod_{r=1}^{T} c_r \not\to 0$, convergence already fails because the initial bias $w_0 - w^*$ remains. Otherwise,

$$\sum_{s=1}^{T} \beta_{s,T} \to 1.$$

By Cauchy's inequality,

$$\sum_{s=1}^{T} \beta_{s,T}^2 \geq \frac{(1 - \prod_{r=1}^{T} c_r)^2}{T}.$$

Therefore, the contribution of the adversarial non-shifted noise to the final risk satisfies

$$\mathbb{E}[(w_T - w^*)^2] \geq \sum_{s=1}^{T} \beta_{s,T}^2 (\sigma^2 + M_T) \geq M_T \sum_{s=1}^{T} \beta_{s,T}^2 \geq M_T \frac{(1 - \prod_{r=1}^{T} c_r)^2}{T}.$$

Since $M_T = \Omega(T)$ and $\prod_{r=1}^{T} c_r \to 0$, the right-hand side is bounded away from zero. Hence convergence cannot be guaranteed under frequent, unbounded, non-shifted attacks. $\square$

## C. Missing details from Section 4

### C.1. Proof of Lemma 4.1

We first present the complete version of Lemma 4.1

**Lemma C.1.** *Define for any $s = 1, \ldots, t$:*

$$Q_s := \nabla_{ww}^2 \mathcal{L}(w_s^*, Z_s), \; G_s := \nabla_{wZ}^2 \mathcal{L}(w_s^*, Z_s), \; S_s := Q_s + H_s.$$

*The detection score computed from* (6) *satisfies*

$$D_t^1(w_t - w_{t-1}) - D_t^2(w_{t-1} - w_{t-2}) = E_t^1(w_t^* - w_{t-1}^*) + E_t^2 \operatorname{vec}(\eta_{t-1}) + E_t^3 \operatorname{vec}(\eta_t)$$
$$+ O(\|w_{t-1} - w_{t-1}^*\|_F^2) + O(\|w_t - w_t^*\|_F^2) + O(\|\eta_t\|_F^2 + \|\eta_{t-1}\|_F^2),$$

*where*

$$
\begin{aligned}
E_t^1 &= D_t^1 A_t + D_t^1 S_t^+ Q_t S_{t-1}^+ Q_{t-1}, \\
E_t^2 &= D_t^1 S_t^+ Q_t S_{t-1}^+ G_{t-1} + D_t^2 S_{t-1}^+ G_{t-1}, \\
E_t^3 &= - D_t^1 S_t^+ G_t, \\
D_t^1 &= (I - B_t)(A_t^+ - (A_t^+ A_t - B_t^+ B_t)(A_t^\top A_t + B_t^\top B_t)^+ A_t^\top), \\
D_t^2 &= (I - B_t)(B_t^+ + (A_t^+ A_t - B_t^+ B_t)(A_t^\top A_t + B_t^\top B_t)^+ B_t^\top), \\
A_t &= S_t^+ Q_t S_{t-1}^+ H_{t-1}, \\
B_t &= S_{t-1}^+ Q_{t-1}.
\end{aligned}
\tag{24}
$$

*Proof of Lemma C.1.* By the CL iteration derived in (20) and (21), we have:

$$w_t - w_{t-1} = S_t^+(Q_t(w_t^* - w_{t-1}) - G_t \operatorname{vec}(\eta_t)) + O(\|w_t - w_t^*\|_F^2) + O(\|\eta_t\|_F^2).$$

and

$$w_{t-1} - w_t^* = S_{t-1}^+(Q_{t-1}(w_{t-1}^* - w_t^*) + H_{t-1}(w_{t-2} - w_t^*) - G_{t-1} \operatorname{vec}(\eta_{t-1}))$$
$$+ O(\|w_{t-1} - w_{t-1}^*\|_F^2) + O(\|\eta_{t-1}\|_F^2).$$

Combining two equations above, we have

$$w_t - w_{t-1} = - A_t(w_{t-2} - w_t^*) + S_t^+ Q_t S_{t-1}^+ Q_{t-1}(w_t^* - w_{t-1}^*) + S_t^+ Q_t S_{t-1}^+ G_{t-1} \operatorname{vec}(\eta_{t-1}) - S_t^+ G_t \operatorname{vec}(\eta_t)$$
$$+ O(\|w_t - w_t^*\|_F^2 + \|w_{t-1} - w_{t-1}^*\|_F^2) + O(\|\eta_t\|_F^2 + \|\eta_{t-1}\|_F^2) \tag{25}$$
$$w_{t-1} - w_{t-2} = - B_t(w_{t-2} - w_{t-1}^*) - S_{t-1}^+ G_{t-1} \operatorname{vec}(\eta_{t-1}) + O(\|w_{t-1} - w_{t-1}^*\|_F^2) + O(\|\eta_{t-1}\|_F^2) \tag{26}$$

The $w_t - w_{t-1}$ and $w_{t-1} - w_{t-2}$ include $A_t w_{t-2}$ and $B_t w_{t-2}$ respectively. Our goal is to find matrices $D_t^1$ and $D_t^2$ such that $D_t^1 A_t = D_t^2 B_t$, thereby removing the component involving $w_{t-2}$. To determine $D_t^1$ and $D_t^2$ in the common subspace of $A_t$ and $B_t$ while retaining as much information as possible regarding the deviations in tasks $t - 1$ and $t$, we formulate

$$\min_{\bar{D}_t^1, \bar{D}_t^2} \|\bar{D}_t^1 - A_t^+\|_F^2 + \|\bar{D}_t^2 - B_t^+\|_F^2, \text{ subject to } \bar{D}_t^1 A_t = \bar{D}_t^2 B_t,$$

where $\| \cdot \|_F$ denotes the Frobenius norm. The closed-form solutions to this problem are

$$\bar{D}_t^1 = A_t^+ - (A_t^+ A_t - B_t^+ B_t)(A_t^\top A_t + B_t^\top B_t)^+ A_t^\top,$$
$$\bar{D}_t^2 = B_t^+ + (A_t^+ A_t - B_t^+ B_t)(A_t^\top A_t + B_t^\top B_t)^+ B_t^\top.$$

We then define the normalized matrices

$$D_t^1 = (I - B_t)\bar{D}_t^1, \qquad D_t^2 = (I - B_t)\bar{D}_t^2.$$

Substituting $D_t^1$ and $D_t^2$ into $D_t^1(w_t - w_{t-1}) - D_t^2(w_{t-1} - w_{t-2})$, and using $D_t^1 A_t = D_t^2 B_t$, we obtain the claimed expression.

$\square$

## C.2. Proof of Lemma 4.3 and Corollary C.2

We first show the following Corollary of Lemma 4.3.

**Corollary C.2.** *If all coordinates of the noise components $\varepsilon_t^{i,j}$ are sub-Gaussian that satisfy*

$$\sup_{p \geq 1} p^{-1/2} (\mathbb{E}(|\varepsilon_t^{i,j}|^p)^{1/p}) \leq \kappa,$$

*the detection score $d_t$ satisfies $d_t \leq \theta_t$ for all $t \in [T]$ with probability at least $1 - \epsilon$ with the tightened threshold:*

$$\theta_t = \sqrt{\sigma^2 \mathrm{tr}\left((E_t^2)^\top E_t^2\right) + \sigma^2 \mathrm{tr}\left((E_t^3)^\top E_t^3\right) + \max\left\{\kappa^2 \|R_t^\top R_t\|_F \sqrt{\frac{1}{c} \log \frac{2T}{\epsilon}}, \frac{\kappa^2 \|R_t^\top R_t\|_2}{c} \log \frac{2T}{\epsilon}\right\}},$$

*where $R_t = [E_t^2 \ E_t^3]$, and $c > 0$ is a universal constant.*

*Proof.* Under Assumption 4.2, we vectorize the parameter and responses as

$$\mathbf{w}_t = \mathrm{vec}(w_t), \qquad \mathbf{w}^* = \mathrm{vec}(w^*), \qquad \mathbf{y}_t = \mathrm{vec}(Y_t), \qquad \mathbf{e}_t = \mathrm{vec}(\varepsilon_t).$$

Let

$$\mathbf{X}_t = I_{p_2} \otimes X_t, \qquad Q_t = \frac{\mathbf{X}_t^\top \mathbf{X}_t}{n_t}, \qquad G_t = -\frac{\mathbf{X}_t^\top}{n_t}, \qquad S_t = Q_t + H_t.$$

Then the linear model can be written as

$$\mathbf{y}_t = \mathbf{X}_t \mathbf{w}^* + \mathbf{e}_t.$$

For benign task $t$, the regularization-based update satisfies

$$\mathbf{w}_t = \arg\min_{\mathbf{w}} \left\{ \frac{1}{2n_t} \|\mathbf{X}_t \mathbf{w} - \mathbf{y}_t\|_2^2 + \frac{1}{2} \|\mathbf{w} - \mathbf{w}_{t-1}\|_{H_t}^2 \right\}.$$

By the first-order optimality condition,

$$Q_t(\mathbf{w}_t - \mathbf{w}^*) + G_t \mathbf{e}_t + H_t(\mathbf{w}_t - \mathbf{w}_{t-1}) = 0.$$

Then, we obtain the recursion

$$\mathbf{w}_t - \mathbf{w}^* = S_t^+ \left( H_t(\mathbf{w}_{t-1} - \mathbf{w}^*) - G_t \mathbf{e}_t \right). \tag{27}$$

Combining $\mathbf{w}_t - \mathbf{w}^*$ and $\mathbf{w}_{t-1} - \mathbf{w}^*$ gives,

$$\mathbf{w}_t - \mathbf{w}_{t-1} = -S_t^+ Q_t(\mathbf{w}_{t-1} - \mathbf{w}^*) - S_t^+ G_t \mathbf{e}_t.$$

Applying the same relation to task $t - 1$ gives

$$\mathbf{w}_{t-1} - \mathbf{w}^* = S_{t-1}^+ \left( H_{t-1}(\mathbf{w}_{t-2} - \mathbf{w}^*) - G_{t-1}\mathbf{e}_{t-1} \right),$$

and hence

$$\mathbf{w}_t - \mathbf{w}_{t-1} = - S_t^+ Q_t S_{t-1}^+ H_{t-1}(\mathbf{w}_{t-2} - \mathbf{w}^*) + S_t^+ Q_t S_{t-1}^+ G_{t-1}\mathbf{e}_{t-1} - S_t^+ G_t \mathbf{e}_t.$$

Also,

$$\mathbf{w}_{t-1} - \mathbf{w}_{t-2} = -S_{t-1}^+ Q_{t-1}(\mathbf{w}_{t-2} - \mathbf{w}^*) - S_{t-1}^+ G_{t-1}\mathbf{e}_{t-1}.$$

Recall that

$$A_t = S_t^+ Q_t S_{t-1}^+ H_{t-1}, \qquad B_t = S_{t-1}^+ Q_{t-1},$$

and that the construction of $D_t^1$ and $D_t^2$ ensures

$$D_t^1 A_t = D_t^2 B_t.$$

Therefore, the historical term involving $\mathbf{w}_{t-2} - \mathbf{w}^*$ cancels in

$$D_t^1(\mathbf{w}_t - \mathbf{w}_{t-1}) - D_t^2(\mathbf{w}_{t-1} - \mathbf{w}_{t-2}),$$

which yields

$$D_t^1(\mathbf{w}_t - \mathbf{w}_{t-1}) - D_t^2(\mathbf{w}_{t-1} - \mathbf{w}_{t-2}) = E_t^2 \mathbf{e}_{t-1} + E_t^3 \mathbf{e}_t,$$

where

$$E_t^2 = D_t^1 S_t^+ Q_t S_{t-1}^+ G_{t-1} + D_t^2 S_{t-1}^+ G_{t-1}, \qquad E_t^3 = -D_t^1 S_t^+ G_t.$$

These are the linear-regression specializations of the matrices defined in Appendix C.1.

Thus, when tasks $t-1$ and $t$ are benign,

$$d_t^2 = \left\| E_t^2 \mathbf{e}_{t-1} + E_t^3 \mathbf{e}_t \right\|_2^2.$$

Conditioned on the feature matrices, $\mathbf{e}_{t-1}$ and $\mathbf{e}_t$ are independent, zero-mean, and satisfy

$$\mathbb{E}[\mathbf{e}_s \mathbf{e}_s^\top \mid X_s] = \sigma^2 I.$$

Therefore, the cross term vanishes and

$$\mathbb{E}[d_t^2 \mid X_{t-1}, X_t] = \sigma^2 \operatorname{tr}\left( (E_t^2)^\top E_t^2 \right) + \sigma^2 \operatorname{tr}\left( (E_t^3)^\top E_t^3 \right).$$

By Markov's inequality,

$$\mathbb{P}\left( d_t^2 > \frac{T}{\epsilon} \sigma^2 \left[ \operatorname{tr}\left( (E_t^2)^\top E_t^2 \right) + \operatorname{tr}\left( (E_t^3)^\top E_t^3 \right) \right] \right) \le \frac{\epsilon}{T}.$$

Taking a union bound over all $t \in [T]$ for which $d_t$ is computed gives, with probability at least $1 - \epsilon$,

$$d_t \le \sqrt{\frac{\sigma^2 T}{\epsilon} \left[ \operatorname{tr}\left( (E_t^2)^\top E_t^2 \right) + \operatorname{tr}\left( (E_t^3)^\top E_t^3 \right) \right]}$$

simultaneously for all such $t$. This proves Lemma 4.3.

Let

$$\xi_t = \begin{bmatrix} \mathbf{e}_{t-1} \\ \mathbf{e}_t \end{bmatrix}, \qquad R_t = [E_t^2 \ E_t^3].$$

Then $d_t^2 = \|R_t \xi_t\|^2 = \xi_t^\top R_t^\top R_t \xi_t$. When the noise $\mathbf{e_t}$ and $\mathbf{e_{t-1}}$ are sub-Gaussian, we have the following Hanson-Wright inequality:

$$\mathbb{P}\left( d_t^2 \le \mathbb{E}[d_t^2] + \max\left\{ \sqrt{\frac{\kappa^4 \|R_t^\top R_t\|_F^2}{c} \log \frac{2T}{\epsilon}}, \frac{\kappa^2 \|R_t^\top R_t\|_2}{c} \log \frac{2T}{\epsilon} \right\} \right) \ge 1 - \frac{\epsilon}{T},$$

where $c$ is a universal constant. A union bound over $t \in [T]$ gives Corollary C.2.  $\square$

### C.3. Proof of Theorem 4.4

We first separate detected and undetected attacks. If an attack is detected, Alg. 1 rolls the model back and discards the poisoned update, so its adversarial bias does not enter the subsequent recursion. Since the attack is infrequent, discarding these finitely many updates does not change the asymptotic convergence rate. Hence the only attacks contributing to the non-benign term in the final recursion are undetected attacks. We therefore focus on such attacks without loss of generality.

We show that any attack that remains undetected under the online information structure of CL must satisfy

$$\left\| \left( \frac{\mathbf{X}_t^\top \mathbf{X}_t}{n_t} \right)^+ \frac{\mathbf{X}_t^\top}{n_t} (\mathbf{n}_t + \mathbf{e}_t) \right\|_2^2 \le \frac{\sigma^2 T}{\epsilon} \operatorname{tr}\left( \left( \mathbf{X}_t^\top \mathbf{X}_t \right)^+ \right). \tag{28}$$

To see this, consider an adversarial perturbation injected at task $t-1$. By extending the derivation in Appendix C.2, its contribution to the next verification score $d_t$ is through the term $E_t^2(\mathbf{n}_{t-1} + \mathbf{e}_{t-1})$. At the time of choosing $\mathbf{n}_{t-1}$, the adversary has observed the history and the current task, but not the future feature matrix $\mathbf{X}_t$. Therefore, an attack that can remain undetected for all admissible future task realizations must in particular remain undetected under the admissible realization whose row space gives the largest effective response in the T2T projection.

For this worst-aligned realization, Lemma C.3 shows that, $A_t^+ A_t = B_t^+ B_t$ and the oblique projection factor $(I - B_t)B_t^+ B_t(I - B_t)^{-1}$ acts as the identity on the relevant row space, and the task-$(t-1)$ component of the score reduces, up to sign, to

$$\left(\mathbf{X}_{t-1}^\top \mathbf{X}_{t-1}\right)^+ \mathbf{X}_{t-1}^\top (\mathbf{n}_{t-1} + \mathbf{e}_{t-1}).$$

If (28) were violated for task $t-1$, then under this admissible future realization the resulting T2T score can exceed the threshold $\theta_t$ with high probability, and the attack would be detected and discarded by Alg. 1. Hence every attack that is not discarded, and thus can enter the final recursion below, must satisfy (28).

Then, extending the recursion in (27) to account for the attack $\mathbf{n}_t$ added to $\mathbf{y}_t$, we obtain

$$\mathbf{w}_t - \mathbf{w}^* = \left(\frac{\mathbf{X}_t^\top \mathbf{X}_t}{n_t} + H_t\right)^{-1} \left(\frac{\mathbf{X}_t^\top (\mathbf{e}_t + \mathbf{n}_t)}{n_t} + H_t(\mathbf{w}_{t-1} - \mathbf{w}^*)\right), \quad t = 1, \dots, T,$$

where $\mathbf{n}_t = \mathbf{0}$ if $t$ is not attacked. Substituting $H_t$ from (9) gives

$$\mathbf{w}_t - \mathbf{w}^* = \left(\frac{\sigma^2}{W}I + \sum_{s=1}^{t} \mathbf{X}_s^\top \mathbf{X}_s\right)^{-1} \left(\mathbf{X}_t^\top (\mathbf{e}_t + \mathbf{n}_t) + \left(\frac{\sigma^2}{W}I + \sum_{s=1}^{t-1} \mathbf{X}_s^\top \mathbf{X}_s\right)(\mathbf{w}_{t-1} - \mathbf{w}^*)\right).$$

Unrolling the recursion yields

$$\mathbf{w}_T - \mathbf{w}^* = \left(\frac{\sigma^2}{W}I + \sum_{s=1}^{T} \mathbf{X}_s^\top \mathbf{X}_s\right)^{-1} \left(\sum_{s=1}^{T} \mathbf{X}_s^\top (\mathbf{e}_s + \mathbf{n}_s) + \frac{\sigma^2}{W}I(\mathbf{w}_0 - \mathbf{w}^*)\right).$$

Let $\mathcal{K} = \{t_1, \dots, t_K\}$ be the set of attacked tasks, and define

$$\tilde{\mathbf{X}}_{T+1} := \frac{\sigma^2}{W}I + \sum_{s=1}^{T} \mathbf{X}_s^\top \mathbf{X}_s.$$

We choose $w_0 = \mathbf{0}$ in the algorithm. The final excess risk satisfies

$$\mathbb{E}\left[\|\mathbf{w}_T - \mathbf{w}^*\|^2\right] = \mathbb{E}\left[\left\|\tilde{\mathbf{X}}_{T+1}^{-1}\left(\frac{\sigma^2}{W}(-\mathbf{w}^*) + \sum_{s \notin \mathcal{K}} \mathbf{X}_s^\top \mathbf{e}_s + \sum_{k=1}^{K} \mathbf{X}_{t_k}^\top (\mathbf{n}_{t_k} + \mathbf{e}_{t_k})\right)\right\|^2\right]$$

$$\leq \sigma^2 \operatorname{tr}\left(\tilde{\mathbf{X}}_{T+1}^{-1}\right) + \mathbb{E}\left[K \sum_{k=1}^{K} \left\|\tilde{\mathbf{X}}_{T+1}^{-1} \mathbf{X}_{t_k}^\top (\mathbf{n}_{t_k} + \mathbf{e}_{t_k})\right\|^2\right], \tag{29}$$

where we only take expectation over the term of $\left\|\tilde{\mathbf{X}}_{T+1}^{-1}\left(\frac{\sigma^2}{W}(-\mathbf{w}^*) + \sum_{s \notin \mathcal{K}} \mathbf{X}_s^\top \mathbf{e}_s\right)\right\|^2$ by the following:

$$\mathbb{E}\left[\left\|\tilde{\mathbf{X}}_{T+1}^{-1}\left(\frac{\sigma^2}{W}(-\mathbf{w}^*) + \sum_{s \notin \mathcal{K}} \mathbf{X}_s^\top \mathbf{e}_s\right)\right\|^2\right] = \left\|\tilde{\mathbf{X}}_{T+1}^{-1} \frac{\sigma^2}{W}(-\mathbf{w}^*)\right\|^2 + \sigma^2 \operatorname{tr}\left(\tilde{\mathbf{X}}_{T+1}^{-1}\left(\sum_{s \notin \mathcal{K}} \mathbf{X}_s^\top \mathbf{X}_s\right)\tilde{\mathbf{X}}_{T+1}^{-1}\right)$$

$$\leq \operatorname{tr}\left(\tilde{\mathbf{X}}_{T+1}^{-1}\left(\frac{\sigma^4}{W}I + \sigma^2 \sum_{s \notin \mathcal{K}} \mathbf{X}_s^\top \mathbf{X}_s\right)\tilde{\mathbf{X}}_{T+1}^{-1}\right)$$

$$\leq \operatorname{tr}\left(\tilde{\mathbf{X}}_{T+1}^{-1}\left(\frac{\sigma^4}{W}I + \sigma^2 \sum_{s=1}^{T} \mathbf{X}_s^\top \mathbf{X}_s\right)\tilde{\mathbf{X}}_{T+1}^{-1}\right)$$

$$= \sigma^2 \operatorname{tr}\left(\tilde{\mathbf{X}}_{T+1}^{-1}\right), \tag{30}$$

where we used $\|\mathbf{w}^*\|^2 \leq W$ and

$$\frac{\sigma^4}{W}I + \sigma^2 \sum_{s=1}^{T} \mathbf{X}_s^\top \mathbf{X}_s = \sigma^2 \tilde{\mathbf{X}}_{T+1}.$$

For each attacked task $t_k$, we further have

$$\left\|\tilde{\mathbf{X}}_{T+1}^{-1}\mathbf{X}_{t_k}^\top(\mathbf{n}_{t_k} + \mathbf{e}_{t_k})\right\|^2 = \left\|\tilde{\mathbf{X}}_{T+1}^{-1}(\mathbf{X}_{t_k}^\top\mathbf{X}_{t_k})(\mathbf{X}_{t_k}^\top\mathbf{X}_{t_k})^+\mathbf{X}_{t_k}^\top(\mathbf{n}_{t_k} + \mathbf{e}_{t_k})\right\|^2$$

$$\leq \left\|\tilde{\mathbf{X}}_{T+1}^{-1}(\mathbf{X}_{t_k}^\top\mathbf{X}_{t_k})\right\|_2^2 \left\|(\mathbf{X}_{t_k}^\top\mathbf{X}_{t_k})^+\mathbf{X}_{t_k}^\top(\mathbf{n}_{t_k} + \mathbf{e}_{t_k})\right\|^2$$

$$\leq \left\|\tilde{\mathbf{X}}_{T+1}^{-1}(\mathbf{X}_{t_k}^\top\mathbf{X}_{t_k})\right\|_2^2 \frac{\sigma^2 T}{\epsilon}\operatorname{tr}\left(\left(\mathbf{X}_{t_k}^\top\mathbf{X}_{t_k}\right)^+\right). \tag{31}$$

Combining (29), (30), and (31), we obtain

$$\mathbb{E}[\|\mathbf{w}_T - \mathbf{w}^*\|^2] \leq \sigma^2\operatorname{tr}\left(\tilde{\mathbf{X}}_{T+1}^{-1}\right) + K\sum_{k=1}^{K}\left\|\tilde{\mathbf{X}}_{T+1}^{-1}(\mathbf{X}_{t_k}^\top\mathbf{X}_{t_k})\right\|_2^2 \frac{\sigma^2 T}{\epsilon}\operatorname{tr}\left(\left(\mathbf{X}_{t_k}^\top\mathbf{X}_{t_k}\right)^+\right), \tag{32}$$

where the last inequality follows from the undetected-task constraint in (28).

When the tasks are informative, the eigenvalues of $\tilde{\mathbf{X}}_{T+1}$ grow on the order of $\Theta(T)$. Since the eigenvalues of each $\mathbf{X}_{t_k}^\top\mathbf{X}_{t_k}$ are uniformly bounded, the bound above yields the convergence rate $O(K^2/(T\epsilon))$.

**Lemma C.3.** *For $E_t^2$ defined in (24), under Assumption 4.2, suppose $H_t$ is chosen as in (9) and $A_t^+A_t = B_t^+B_t$. Let $\mathbf{n}_{t-1} := \operatorname{vec}(\eta_{t-1})$ and $\mathbf{e}_{t-1} := \operatorname{vec}(\varepsilon_{t-1})$. Then*

$$E_t^2(\mathbf{e}_{t-1} + \mathbf{n}_{t-1}) = -(I - B_t)B_t^+B_t(I - B_t)^{-1}(\mathbf{X}_{t-1}^\top\mathbf{X}_{t-1})^+\mathbf{X}_{t-1}^\top(\mathbf{e}_{t-1} + \mathbf{n}_{t-1}),$$

*up to the sign convention in $G_{t-1}$.*

*Proof of Lemma C.3.* Firstly, note that $H_t$ in (9) is full rank for any $t$, thus $S_t$ defined in Lemma C.1 is also full rank. Since $A_t^+A_t = B_t^+B_t$, we have

$$D_t^1 = (I - B_t)A_t^+, \qquad D_t^2 = (I - B_t)B_t^+.$$

Note that $S_s$ and $H_s$ are invertible for all $s$. Therefore, $E_t^2$ in Lemma C.1 simplifies as follows:

$$\begin{aligned}
E_t^2 &= \left[D_t^1 S_t^{-1}Q_t + D_t^2\right]S_{t-1}^{-1}G_{t-1}\\
&= (I - B_t)\left(A_t^+ S_t^{-1}Q_t + B_t^+\right)S_{t-1}^{-1}Q_{t-1}Q_{t-1}^+G_{t-1}\\
&= (I - B_t)\left((S_t^{-1}Q_tS_{t-1}^{-1}H_{t-1})^+S_t^{-1}Q_t + (S_{t-1}^{-1}Q_{t-1})^+\right)S_{t-1}^{-1}Q_{t-1}Q_{t-1}^+G_{t-1}\\
&= (I - B_t)\left((S_t^{-1}Q_tS_{t-1}^{-1}H_{t-1})^+S_t^{-1}Q_tS_{t-1}^{-1}H_{t-1}H_{t-1}^{-1}S_{t-1} + (S_{t-1}^{-1}Q_{t-1})^+\right)S_{t-1}^{-1}Q_{t-1}Q_{t-1}^+G_{t-1}\\
&= (I - B_t)\left(I_1 H_{t-1}^{-1}S_{t-1}S_{t-1}^{-1}Q_{t-1} + I_2\right)Q_{t-1}^+G_{t-1}\\
&= (I - B_t)\left(I_1 H_{t-1}^{-1}Q_{t-1} + I_2\right)Q_{t-1}^+G_{t-1},
\end{aligned}$$

where we use $G_{t-1} = Q_{t-1}Q_{t-1}^+G_{t-1}$ in the second equation because $\mathbf{X}_{t-1}^\top$ is in the same space as $\mathbf{X}_{t-1}^\top\mathbf{X}_{t-1}$, and

$$I_1 = A_t^+A_t, \qquad I_2 = B_t^+B_t.$$

Since $I_1 = I_2$, we have

$$\begin{aligned}
(I - B_t)\left(I_1 H_{t-1}^{-1}Q_{t-1} + I_2\right)Q_{t-1}^+G_{t-1} &= (I - B_t)I_1 H_{t-1}^{-1}S_{t-1}Q_{t-1}^+G_{t-1}\\
&= (I - B_t)I_1(I - B_t)^{-1}Q_{t-1}^+G_{t-1}\\
&= (I - B_t)B_t^+B_t(I - B_t)^{-1}Q_{t-1}^+G_{t-1},
\end{aligned}$$

where the second equality follows from $I - B_t = S_{t-1}^{-1}H_{t-1}$. Under Assumption 4.2, $Q_{t-1}^+G_{t-1}$ equals

$$-\left(\mathbf{X}_{t-1}^\top\mathbf{X}_{t-1}\right)^+\mathbf{X}_{t-1}^\top.$$

Multiplying both sides by $\mathbf{e}_{t-1} + \mathbf{n}_{t-1}$ completes the proof.

$\square$

# D. Missing details of Section 5

## D.1. Extension of (11) under Assumptions 4.2 and 5.1

*Derivation of* (12) *and* (13): For simplicity, in this subsection we drop the boldface notation and write $w_t \leftarrow \mathbf{w}_t$, $y_t \leftarrow \mathbf{y}_t$, $X_t \leftarrow \mathbf{X}_t$, $\tilde{\eta}_t \leftarrow \mathbf{n}_t$, and $\varepsilon_t \leftarrow \mathbf{e}_t$, where these quantities are defined in Appendix C.2, and $\mathbf{n}_t$ contains only stochastic terms with conditional zero-mean. Then, extending the recursion in (27) to account for the non-shifted attack $\tilde{\eta}_t$ added to $y_t$, we obtain

$$w_t - w^* = S_t^{-1} \left( \frac{X_t^\top (\varepsilon_t + \tilde{\eta}_t)}{n_t} + H_t(w_{t-1} - w_*) \right), \quad t = 1, \ldots, T, \tag{33}$$

where

$$S_t = \frac{X_t^\top X_t}{n_t} + H_t.$$

Since the attack is non-shifted and the inherent noise satisfies $\mathbb{E}[\varepsilon_t] = 0$ and $\mathbb{E}[\varepsilon_t \varepsilon_t^\top] = \sigma^2 I$, the mean recursion is

$$\mathbb{E}[w_t - w_*] = S_t^{-1} H_t \mathbb{E}[w_{t-1} - w_*]. \tag{34}$$

Moreover, since the current noise terms are conditionally uncorrelated with $w_{t-1} - w_*$, the covariance recursion is

$$\mathrm{Cov}(w_t - w_*) = S_t^{-1} \frac{X_t^\top}{n_t} \mathrm{Cov}(\tilde{\eta}_t) \frac{X_t}{n_t} S_t^{-1} + \sigma^2 S_t^{-1} \frac{X_t^\top X_t}{n_t^2} S_t^{-1} + S_t^{-1} H_t \mathrm{Cov}(w_{t-1} - w_*) H_t^\top S_t^{-1}, \tag{35}$$

this proves (13). Therefore,

$$\begin{aligned}
\mathcal{R}(w_t) & \tag{36} \\
&= \mathbb{E}[\|w_t - w_*\|_2^2] \\
&= \|\mathbb{E}[w_t - w_*]\|_2^2 + \mathrm{tr}(\mathrm{Cov}(w_t - w_*)) \\
&= \left\| S_t^{-1} H_t \mathbb{E}[w_{t-1} - w_*] \right\|_2^2 + \mathrm{tr}\left( S_t^{-1} \frac{X_t^\top}{n_t} \mathrm{Cov}(\tilde{\eta}_t) \frac{X_t}{n_t} S_t^{-1} \right) + \mathrm{tr}\left( \sigma^2 S_t^{-1} \frac{X_t^\top X_t}{n_t^2} S_t^{-1} \right) \\
&\quad + \mathrm{tr}\left( S_t^{-1} H_t \mathrm{Cov}(w_{t-1} - w_*) H_t^\top S_t^{-1} \right).
\end{aligned} \tag{37}$$

Here we used $S_t^\top = S_t$, since $H_t$ and $X_t^\top X_t$ are symmetric.

Thus, at each task $t$, the defender can choose $H_t$ against the worst-case non-shifted attack covariance by solving $\min_{H_t} \max_{\mathrm{Cov}(\tilde{\eta}_t) \succeq 0} \mathcal{R}(w_t)$ as

$$\begin{aligned}
\min_{H_t} \max_{\mathrm{Cov}(\tilde{\eta}_t) \succeq 0} & \left\| S_t^{-1} H_t \mathbb{E}[w_{t-1} - w_*] \right\|_2^2 + \mathrm{tr}\left( S_t^{-1} \frac{X_t^\top}{n_t} \mathrm{Cov}(\tilde{\eta}_t) \frac{X_t}{n_t} S_t^{-1} \right) \\
& + \mathrm{tr}\left( \sigma^2 S_t^{-1} \frac{X_t^\top X_t}{n_t^2} S_t^{-1} \right) + \mathrm{tr}\left( S_t^{-1} H_t \mathrm{Cov}(w_{t-1} - w_*) H_t^\top S_t^{-1} \right) \\
\text{s.t. } & \mathrm{tr}(\mathrm{Cov}(\tilde{\eta}_t)) \leq M.
\end{aligned} \tag{38}$$

This gives the optimization problem in (12). Since $w = 0$, $\mathrm{Cov}(w_0 - w_*) = 0$ and $\|\mathbb{E}[w_0 - w_*]\|_2^2 = \|w_*\|_2^2 \leq W$.

Under Assumption 5.1, write

$$\frac{X_t^\top X_t}{n_t} = U\Gamma_t U^\top, \qquad \Gamma_t = \mathrm{diag}(\gamma_1^t, \ldots, \gamma_p^t).$$

For each direction $j$ with $\gamma_j^t > 0$, define

$$v_j^t := \frac{X_t u_j}{\sqrt{n_t \gamma_j^t}},$$

so that $\|v_j^t\|_2 = 1$ and

$$X_t u_j = \sqrt{n_t \gamma_j^t}\, v_j^t.$$

For directions with $\gamma_j^t = 0$, we set the corresponding noise-sensitivity term to zero.

Projecting the recursion in (33) onto $u_j$ gives, for $\gamma_j^t > 0$,

$$u_j^\top (w_t - w_*) = \frac{1}{\lambda_j^t + \gamma_j^t} \left( \frac{\sqrt{\gamma_j^t}}{\sqrt{n_t}} (v_j^t)^\top (\varepsilon_t + \eta_t) + \lambda_j^t u_j^\top (w_{t-1} - w_*) \right). \tag{39}$$

For $\gamma_j^t = 0$, the same formula holds. Since both the inherent noise and the non-shifted attack have zero mean and are conditionally uncorrelated with $w_{t-1} - w_*$, the cross terms vanish. Therefore,

$$\mathcal{R}_j^t := \mathbb{E}\left[ (u_j^\top (w_t - w_*))^2 \right] = \left( \frac{\lambda_j^t}{\lambda_j^t + \gamma_j^t} \right)^2 \mathcal{R}_j^{t-1} + \frac{\gamma_j^t \left( \sigma^2 + (v_j^t)^\top \mathrm{Cov}(\tilde{\eta}_t) v_j^t \right)}{n_t (\lambda_j^t + \gamma_j^t)^2}. \tag{40}$$

Thus, the minimax problem in (12) becomes

$$\min_{\lambda_1^t, \dots, \lambda_p^t} \max_{\mathrm{Cov}(\tilde{\eta}_t) \succeq 0} \quad \sum_{j=1}^p \left[ \left( \frac{\lambda_j^t}{\lambda_j^t + \gamma_j^t} \right)^2 \mathcal{R}_j^{t-1} + \frac{\gamma_j^t \left( \sigma^2 + (v_j^t)^\top \mathrm{Cov}(\tilde{\eta}_t) v_j^t \right)}{n_t (\lambda_j^t + \gamma_j^t)^2} \right]$$

$$\text{s.t.} \quad \mathrm{tr}\left( \mathrm{Cov}(\tilde{\eta}_t) \right) \leq M. \tag{41}$$

For fixed $\{\lambda_j^t\}_{j=1}^p$, the adversary's inner problem is linear in $\mathrm{Cov}(\tilde{\eta}_t)$. Since $\{v_j^t\}$ are orthonormal, the optimal adversary allocates its entire covariance budget to the most sensitive direction:

$$j^\star \in \arg\max_{j \in [p]} \frac{\gamma_j^t}{n_t (\lambda_j^t + \gamma_j^t)^2}, \qquad \mathrm{Cov}(\tilde{\eta}_t) = M v_{j^\star}^t (v_{j^\star}^t)^\top.$$

Substituting this worst-case covariance into (41), we obtain

$$\min_{\lambda_1^t, \dots, \lambda_p^t} \max_{j \in [p]} \frac{M \gamma_j^t}{n_t (\lambda_j^t + \gamma_j^t)^2} + \sum_{j=1}^p \left[ \left( \frac{\lambda_j^t}{\lambda_j^t + \gamma_j^t} \right)^2 \mathcal{R}_j^{t-1} + \frac{\gamma_j^t \sigma^2}{n_t (\lambda_j^t + \gamma_j^t)^2} \right], \tag{42}$$

$$\text{s.t.} \quad \mathcal{R}_j^t = \left( \frac{\lambda_j^t}{\lambda_j^t + \gamma_j^t} \right)^2 \mathcal{R}_j^{t-1} + \frac{\gamma_j^t \left( \sigma^2 + (v_j^t)^\top \mathrm{Cov}(\tilde{\eta}_t) v_j^t \right)}{n_t (\lambda_j^t + \gamma_j^t)^2},$$

$$\mathcal{R}_j^0 \leq W, \qquad j = 1, \dots, p. \tag{43}$$

### D.2. Proof of Proposition 5.2

To prove the closed-form solution of problem (43), we take

$$c_j^t = \frac{\lambda_j^t}{\lambda_j^t + \gamma_j^t}.$$

We first handle two degenerate cases. If $\gamma_j^t = 0$, then the current task has no signal in direction $j$, and the attack sensitivity in this direction is zero. Thus this direction does not enter the inner maximization. For any $\lambda_j^t > 0$, we already have $c_j^t = 1$ and $\mathcal{R}_j^t = \mathcal{R}_j^{t-1}$. If $\mathcal{R}_j^{t-1} = 0$ and $\gamma_j^t > 0$, then the objective contribution and the attack sensitivity in direction $j$ are minimized by taking $c_j^t = 1$, equivalently $\lambda_j^t = +\infty$ in the limiting sense.

Therefore, it remains to solve the non-degenerate problem over

$$\mathcal{I}_t^+ := \{ j \in [p] : \gamma_j^t > 0, \ \mathcal{R}_j^{t-1} > 0 \}.$$

If $\mathcal{I}_t^+ = \emptyset$, the above degenerate choices are optimal. In the following, assume $\mathcal{I}_t^+ \neq \emptyset$. After omitting terms independent of the optimization over $\mathcal{I}_t^+$, problem (43) becomes

$$\min_{\{c_j^t\}_{j \in \mathcal{I}_t^+}} \max_{j \in \mathcal{I}_t^+} \frac{(1 - c_j^t)^2}{\gamma_j^t n_t} M + \sum_{j \in \mathcal{I}_t^+} \left[ (c_j^t)^2 \mathcal{R}_j^{t-1} + \frac{(1 - c_j^t)^2 \sigma^2}{n_t \gamma_j^t} \right].$$

We introduce an auxiliary variable $z$ to handle the maximum term. The problem can be reformulated as

$$\min_{\{c_j^t\}_{j \in \mathcal{I}_t^+}, z} \quad \sum_{j \in \mathcal{I}_t^+} \left[ (c_j^t)^2 \mathcal{R}_j^{t-1} + \frac{(1 - c_j^t)^2 \sigma^2}{n_t \gamma_j^t} \right] + zM$$

$$\text{s.t.} \quad z \geq \frac{(1 - c_j^t)^2}{\gamma_j^t n_t}, \qquad \forall j \in \mathcal{I}_t^+. \tag{44}$$

This is a convex optimization problem since the objective function is a sum of convex functions of $c_j^t$ and a linear function of $z$, and the feasible set is the epigraph of convex functions.

Let $\chi_j \geq 0$ be the Lagrange multiplier associated with the $j$-th constraint

$$\frac{(1 - c_j^t)^2}{\gamma_j^t n_t} - z \leq 0.$$

The Lagrangian is

$$g(c^t, z, \chi) = \sum_{j \in \mathcal{I}_t^+} \left[ (c_j^t)^2 \mathcal{R}_j^{t-1} + \frac{(1 - c_j^t)^2 \sigma^2}{n_t \gamma_j^t} \right] + zM + \sum_{j \in \mathcal{I}_t^+} \chi_j \left( \frac{(1 - c_j^t)^2}{\gamma_j^t n_t} - z \right).$$

The KKT conditions for optimality are:

- Stationarity:

$$\frac{\partial L}{\partial z} = M - \sum_{j \in \mathcal{I}_t^+} \chi_j = 0 \implies \sum_{j \in \mathcal{I}_t^+} \chi_j = M,$$

$$\frac{\partial L}{\partial c_j^t} = 2 c_j^t \mathcal{R}_j^{t-1} - \frac{2(1 - c_j^t)(\sigma^2 + \chi_j)}{n_t \gamma_j^t} = 0$$

$$\implies c_j^t = \frac{(\sigma^2 + \chi_j)/(n_t \gamma_j^t)}{\mathcal{R}_j^{t-1} + (\sigma^2 + \chi_j)/(n_t \gamma_j^t)}. \tag{45}$$

- Primal feasibility:

$$z \geq \frac{(1 - c_j^t)^2}{\gamma_j^t n_t}, \qquad \forall j \in \mathcal{I}_t^+.$$

- Dual feasibility:

$$\chi_j \geq 0, \qquad \forall j \in \mathcal{I}_t^+.$$

- Complementary slackness:

$$\chi_j \left( \frac{(1 - c_j^t)^2}{\gamma_j^t n_t} - z \right) = 0, \qquad \forall j \in \mathcal{I}_t^+.$$

From complementary slackness, if $\chi_j > 0$, then the corresponding constraint is active:

$$z = \frac{(1 - c_j^t)^2}{\gamma_j^t n_t}.$$

Using (45), we obtain

$$1 - c_j^t = \frac{\mathcal{R}_j^{t-1}}{\mathcal{R}_j^{t-1} + (\sigma^2 + \chi_j)/(n_t \gamma_j^t)}.$$

Substituting this into the active constraint gives

$$z = \frac{1}{\gamma_j^t n_t} \left( \frac{\mathcal{R}_j^{t-1}}{\mathcal{R}_j^{t-1} + (\sigma^2 + \chi_j)/(n_t \gamma_j^t)} \right)^2$$

$$\sqrt{z \gamma_j^t n_t} = \frac{\mathcal{R}_j^{t-1}}{\mathcal{R}_j^{t-1} + (\sigma^2 + \chi_j)/(n_t \gamma_j^t)}$$

$$\chi_j = \frac{\mathcal{R}_j^{t-1} \sqrt{\gamma_j^t n_t}}{\sqrt{z}} - \sigma^2 - \gamma_j^t n_t \mathcal{R}_j^{t-1}. \tag{46}$$

This expression holds for every $j$ with $\chi_j > 0$.

Let $J_t := \{ j \in \mathcal{I}_t^+ : \chi_j > 0 \}$ be the active set in the optimal solution. Using the stationarity condition $\sum_{j \in \mathcal{I}_t^+} \chi_j = M$, summing (46) gives

$$M = \sum_{j \in J_t} \left( \frac{\mathcal{R}_j^{t-1} \sqrt{\gamma_j^t n_t}}{\sqrt{z}} - \sigma^2 - \gamma_j^t n_t \mathcal{R}_j^{t-1} \right)$$

$$\frac{1}{\sqrt{z}} = \frac{M + |J_t| \sigma^2 + \sum_{j \in J_t} \gamma_j^t n_t \mathcal{R}_j^{t-1}}{\sum_{j \in J_t} \mathcal{R}_j^{t-1} \sqrt{\gamma_j^t n_t}}. \tag{47}$$

We next prove that the final active set $J_t$ must be the set identified in (14). Define, for $j \in \mathcal{I}_t^+$,

$$a_j := \mathcal{R}_j^{t-1} \sqrt{\gamma_j^t n_t} > 0, \qquad b_j := \frac{\gamma_j^t n_t \mathcal{R}_j^{t-1} + \sigma^2}{\mathcal{R}_j^{t-1} \sqrt{\gamma_j^t n_t}}.$$

For any candidate active set $J \subseteq \mathcal{I}_t^+$, the KKT conditions in (47) give

$$\frac{1}{\sqrt{z}} = A_J := \frac{M + \sum_{j \in J} a_j b_j}{\sum_{j \in J} a_j}.$$

Moreover, for every $k \in J$,

$$\chi_k = a_k (A_J - b_k).$$

Hence the dual feasibility condition $\chi_k > 0$ for active constraints is equivalent to

$$b_k < A_J, \qquad k \in J.$$

For any inactive index $k \in \mathcal{I}_t^+ \setminus J$, we have $\chi_k = 0$. In this case, the unconstrained optimizer for $(c_k^t)^2 \mathcal{R}_k^{t-1} + \frac{(1-c_k^t)^2 \sigma^2}{n_t \gamma_k^t}$ satisfies

$$\frac{(1 - c_k^t)^2}{\gamma_k^t n_t} = \frac{1}{b_k^2}.$$

Thus the primal feasibility condition

$$z \geq \frac{(1 - c_k^t)^2}{\gamma_k^t n_t}$$

is equivalent to

$$A_J \leq b_k, \qquad k \in \mathcal{I}_t^+ \setminus J.$$

Consequently, $J$ can be the final active set only if

$$b_k < A_J \quad \text{for all } k \in J, \qquad b_k \geq A_J \quad \text{for all } k \in \mathcal{I}_t^+ \setminus J.$$

Now reorder the non-degenerate indices in $\mathcal{I}_t^+$ such that

$$b_1 \leq b_2 \leq \cdots \leq b_{|\mathcal{I}_t^+|}.$$

Degenerate indices with $\gamma_j^t = 0$ or $\mathcal{R}_j^{t-1} = 0$ are treated as having $b_j = +\infty$, so they never enter the active set. Suppose $k \in J$. Then $b_k < A_J$. For any $i < k$, we have $b_i \leq b_k < A_J$. If $i \notin J$, inactive feasibility would require $b_i \geq A_J$, a contradiction. Thus every $i < k$ must also belong to $J$. Therefore, the final active set must be a prefix of the sorted index set:

$$J = \{1, \ldots, m\}$$

for some $m$.

It remains to identify $m$. For a prefix $J_m = \{1, \ldots, m\}$, define

$$A_m := \frac{M + \sum_{k=1}^m a_k b_k}{\sum_{k=1}^m a_k} = \frac{M + m\sigma^2 + \sum_{k=1}^m \gamma_k^t n_t \mathcal{R}_k^{t-1}}{\sum_{k=1}^m \mathcal{R}_k^{t-1} \sqrt{\gamma_k^t n_t}}.$$

The KKT conditions for the prefix $J_m$ require

$$A_m > b_m$$

for the last active index, and

$$A_m \leq b_{m+1}$$

for the first inactive index, with the second condition omitted when $m = |\mathcal{I}_t^+|$. Indeed, if $A_m > b_{m+1}$, then

$$A_{m+1} = \frac{A_m \sum_{k=1}^m a_k + a_{m+1} b_{m+1}}{\sum_{k=1}^{m+1} a_k} > b_{m+1},$$

so $m + 1$ would also satisfy the active condition. Conversely, if $A_m \leq b_{m+1}$, then

$$A_{m+1} = \frac{A_m \sum_{k=1}^m a_k + a_{m+1} b_{m+1}}{\sum_{k=1}^{m+1} a_k} \leq b_{m+1} \leq b_{m+2}.$$

Repeating this argument gives $A_\ell \leq b_\ell$ for all $\ell > m$. Therefore no larger index can satisfy $A_\ell > b_\ell$, and $m$ is exactly the largest index satisfying $A_m > b_m$. Equivalently,

$$m = \max \left\{ j : \frac{M + j\sigma^2 + \sum_{k=1}^j \gamma_k^t n_t \mathcal{R}_k^{t-1}}{\sum_{k=1}^j \mathcal{R}_k^{t-1} \sqrt{\gamma_k^t n_t}} > \frac{\gamma_j^t n_t \mathcal{R}_j^{t-1} + \sigma^2}{\mathcal{R}_j^{t-1} \sqrt{\gamma_j^t n_t}} \right\}.$$

This is precisely the index $j_t$ defined in Proposition 5.2, after sorting the indices by increasing $b_j$. Therefore, the final active set is $J_t = \{1, \ldots, j_t\}$.

Finally, substituting the expression for $\chi_j$ into (45) gives $c_j^t$, and using

$$\lambda_j^t = \frac{c_j^t \gamma_j^t}{1 - c_j^t}$$

yields

$$\lambda_j^t = A_{j_t} \sqrt{\frac{\gamma_j^t}{n_t}} - \gamma_j^t, \qquad j \leq j_t,$$

where

$$A_{j_t} = \frac{M + j_t \sigma^2 + \sum_{k=1}^{j_t} \gamma_k^t n_t \mathcal{R}_k^{t-1}}{\sum_{k=1}^{j_t} \mathcal{R}_k^{t-1} \sqrt{\gamma_k^t n_t}}.$$

For inactive non-degenerate directions $j > j_t$, we have $\chi_j = 0$, and (45) gives

$$\lambda_j^t = \frac{\sigma^2}{n_t \mathcal{R}_j^{t-1}}.$$

For directions with $\mathcal{R}_j^{t-1} = 0$, this formula is understood in the limiting sense as $\lambda_j^t = +\infty$. Directions with $\gamma_j^t = 0$ do not affect the current minimax problem. Any positive $\lambda_j^t$ is optimal for the current task, and we may use the same inactive-direction convention above when $\mathcal{R}_j^{t-1} > 0$. This completes the proof of (15).

## D.3. Proof of Theorem 5.3

**Theorem D.1.** *Suppose the protected set selected by Proposition 5.2 is stationary over time and equals $\{1, \ldots, j_T\}$. For $j \leq j_T$, define*

$$f_j^t := \frac{\mathcal{R}_j^t}{\sum_{k=1}^{j_T} \mathcal{R}_k^t},$$

*and assume $f_j^t \to f_j$ as $t \to \infty$. Assume also that, for $j \leq j_T$, $\tilde{\gamma}_j^t \to \tilde{\gamma}_j^T$ along the stationary problem instance. Then our robust feature defense in (15) satisfies*

$$\mathcal{R}(w_T) \lesssim \frac{M + j_T \sigma^2}{T \left( \sum_{j=1}^{j_T} f_j \sqrt{\tilde{\gamma}_j^T} \right)^2} + \sum_{j=j_T+1}^{p} \frac{\sigma^2}{\sigma^2/W + \sum_{\tau=1}^{T} \tilde{\gamma}_j^\tau}. \tag{48}$$

*For the baseline regularizer $H_t$ in (9), if the maximally sensitive direction is stationary over time, then*

$$\mathcal{R}(w_T) \lesssim M \max_j \frac{\sum_{\tau=1}^{T} \tilde{\gamma}_j^\tau}{\left( \sigma^2/W + \sum_{\tau=1}^{T} \tilde{\gamma}_j^\tau \right)^2} + \sum_{j=1}^{p} \frac{\sigma^2}{\sigma^2/W + \sum_{\tau=1}^{T} \tilde{\gamma}_j^\tau}. \tag{49}$$

*In particular, when $\sum_{\tau=1}^{T} \tilde{\gamma}_j^\tau \sim T\tilde{\gamma}_j^T$, the first term in (49) reduces to*

$$\max_j \frac{M}{T\tilde{\gamma}_j^T}.$$

We prove the theorem under the stationary protected-set regime. Suppose that the protected set selected by Proposition 5.2 is fixed over time and equals $\{1, \ldots, j_T\}$. We assume $j_T \geq 1$; if $j_T = 0$, the protected-set term below is absent and only the attack-free recursion for the inactive directions remains. Recall that

$$\tilde{\gamma}_j^t = \gamma_j^t n_t.$$

For $j \leq j_T$, define the relative error share

$$f_j^t := \frac{\mathcal{R}_j^t}{\sum_{k=1}^{j_T} \mathcal{R}_k^t},$$

and assume $f_j^t \to f_j$ as $t \to \infty$. In the stationary instance, we also write the limiting value of $\tilde{\gamma}_j^t$ as $\tilde{\gamma}_j^T$.

Let

$$\Phi_t := \sum_{j=1}^{j_T} \mathcal{R}_j^t$$

be the aggregate loss over the protected directions. We first derive the recursion for $\Phi_t$. For a fixed task $t$, the scalar minimax objective in (43) can be written as

$$\sum_{j=1}^{p} \mathcal{R}_j^t = \max_j \frac{(1-c_j^t)^2}{\tilde{\gamma}_j^t} M + \sum_{j=1}^{p} \left[ (c_j^t)^2 \mathcal{R}_j^{t-1} + \frac{(1-c_j^t)^2 \sigma^2}{\tilde{\gamma}_j^t} \right], \tag{50}$$

where

$$c_j^t = \frac{\lambda_j^t}{\lambda_j^t + \gamma_j^t}.$$

For protected directions $j \leq j_T$, the KKT condition in Proposition 5.2 makes the constraints active and equalizes the adversarial sensitivity:

$$\frac{(1-c_j^t)^2}{\tilde{\gamma}_j^t} = z_t, \qquad j = 1, \ldots, j_T.$$

Since $0 \leq c_j^t \leq 1$, this gives

$$1 - c_j^t = \sqrt{z_t \tilde{\gamma}_j^t}, \qquad c_j^t = 1 - \sqrt{z_t \tilde{\gamma}_j^t}.$$

From the proof of Proposition 5.2, the active-set KKT condition also gives

$$\frac{1}{\sqrt{z_t}} = \frac{M + j_T \sigma^2 + \sum_{i=1}^{j_T} \tilde{\gamma}_i^t \mathcal{R}_i^{t-1}}{\sum_{i=1}^{j_T} \sqrt{\tilde{\gamma}_i^t} \mathcal{R}_i^{t-1}}.$$

Equivalently,

$$\sqrt{z_t} = \frac{\sum_{i=1}^{j_T} \sqrt{\tilde{\gamma}_i^t} \mathcal{R}_i^{t-1}}{M + j_T \sigma^2 + \sum_{i=1}^{j_T} \tilde{\gamma}_i^t \mathcal{R}_i^{t-1}}.$$

Now sum the risk contributions over $j \le j_T$. Using $c_j^t = 1 - \sqrt{z_t \tilde{\gamma}_j^t}$, we have

$$\begin{aligned}
\Phi_t &= \sum_{j=1}^{j_T} (c_j^t)^2 \mathcal{R}_j^{t-1} + \sum_{j=1}^{j_T} \frac{(1 - c_j^t)^2 \sigma^2}{\tilde{\gamma}_j^t} + M z_t \\
&= \sum_{j=1}^{j_T} \left(1 - \sqrt{z_t \tilde{\gamma}_j^t}\right)^2 \mathcal{R}_j^{t-1} + \sum_{j=1}^{j_T} z_t \sigma^2 + M z_t \\
&= \sum_{j=1}^{j_T} \mathcal{R}_j^{t-1} - 2\sqrt{z_t} \sum_{j=1}^{j_T} \sqrt{\tilde{\gamma}_j^t} \mathcal{R}_j^{t-1} + z_t \sum_{j=1}^{j_T} \tilde{\gamma}_j^t \mathcal{R}_j^{t-1} + j_T z_t \sigma^2 + M z_t \\
&= \Phi_{t-1} - 2\sqrt{z_t} \sum_{j=1}^{j_T} \sqrt{\tilde{\gamma}_j^t} \mathcal{R}_j^{t-1} + z_t \left(M + j_T \sigma^2 + \sum_{j=1}^{j_T} \tilde{\gamma}_j^t \mathcal{R}_j^{t-1}\right).
\end{aligned}$$

Substituting the expression for $\sqrt{z_t}$ into the last display yields

$$\Phi_t = \Phi_{t-1} - \frac{\left(\sum_{j=1}^{j_T} \sqrt{\tilde{\gamma}_j^t} \mathcal{R}_j^{t-1}\right)^2}{M + j_T \sigma^2 + \sum_{j=1}^{j_T} \tilde{\gamma}_j^t \mathcal{R}_j^{t-1}}. \tag{51}$$

Next, rewrite this recursion in terms of the relative shares $f_j^{t-1}$. Since

$$\mathcal{R}_j^{t-1} = f_j^{t-1} \Phi_{t-1},$$

we have

$$\sum_{j=1}^{j_T} \sqrt{\tilde{\gamma}_j^t} \mathcal{R}_j^{t-1} = \Phi_{t-1} \sum_{j=1}^{j_T} f_j^{t-1} \sqrt{\tilde{\gamma}_j^t},$$

and

$$\sum_{j=1}^{j_T} \tilde{\gamma}_j^t \mathcal{R}_j^{t-1} = \Phi_{t-1} \sum_{j=1}^{j_T} f_j^{t-1} \tilde{\gamma}_j^t.$$

Therefore (51) becomes

$$\Phi_t = \Phi_{t-1} - \frac{\Phi_{t-1}^2 \left(\sum_{j=1}^{j_T} f_j^{t-1} \sqrt{\tilde{\gamma}_j^t}\right)^2}{M + j_T \sigma^2 + \Phi_{t-1} \sum_{j=1}^{j_T} f_j^{t-1} \tilde{\gamma}_j^t}. \tag{52}$$

Equivalently,

$$\Phi_t = \Phi_{t-1} \frac{M + j_T \sigma^2 + \Phi_{t-1} \left[\sum_{j=1}^{j_T} f_j^{t-1} \tilde{\gamma}_j^t - \left(\sum_{j=1}^{j_T} f_j^{t-1} \sqrt{\tilde{\gamma}_j^t}\right)^2\right]}{M + j_T \sigma^2 + \Phi_{t-1} \sum_{j=1}^{j_T} f_j^{t-1} \tilde{\gamma}_j^t}. \tag{53}$$

Taking reciprocals in (53), we obtain

$$\frac{1}{\Phi_t} - \frac{1}{\Phi_{t-1}} = \frac{\left(\sum_{j=1}^{j_T} f_j^{t-1}\sqrt{\tilde{\gamma}_j^t}\right)^2}{M + j_T\sigma^2 + \Phi_{t-1}\left[\sum_{j=1}^{j_T} f_j^{t-1}\tilde{\gamma}_j^t - \left(\sum_{j=1}^{j_T} f_j^{t-1}\sqrt{\tilde{\gamma}_j^t}\right)^2\right]}. \tag{54}$$

We now analyze the asymptotic behavior of this recursion. Since $\sum_{j=1}^{j_T} f_j^{t-1} = 1$, Cauchy's inequality gives

$$\left(\sum_{j=1}^{j_T} f_j^{t-1}\sqrt{\tilde{\gamma}_j^t}\right)^2 \leq \sum_{j=1}^{j_T} f_j^{t-1}\tilde{\gamma}_j^t.$$

Hence the bracketed term in the denominator of (54) is nonnegative. It follows from (52) that

$$0 \leq \Phi_t \leq \Phi_{t-1},$$

so $\{\Phi_t\}$ is non-increasing and has a limit, say $\Phi_\infty \geq 0$.

We claim that $\Phi_\infty = 0$. Under the stationarity assumption,

$$\left(\sum_{j=1}^{j_T} f_j^{t-1}\sqrt{\tilde{\gamma}_j^t}\right)^2 \to \left(\sum_{j=1}^{j_T} f_j\sqrt{\tilde{\gamma}_j^T}\right)^2 =: C_\star > 0.$$

The positivity follows because $j_T \geq 1$, $\tilde{\gamma}_j^T > 0$ on the protected set, and $\sum_{j=1}^{j_T} f_j = 1$. The bracketed term in (54) is bounded under the stationary regime. Hence, if $\Phi_\infty > 0$, then the right-hand side of (54) is eventually bounded below by a positive constant. This would imply that $1/\Phi_t \to \infty$, contradicting $\Phi_t \to \Phi_\infty > 0$. Therefore,

$$\Phi_t \to 0.$$

Since $\Phi_{t-1} \to 0$, the denominator in (54) converges to $M + j_T\sigma^2$, and the numerator converges to $C_\star$. Thus

$$\frac{1}{\Phi_t} - \frac{1}{\Phi_{t-1}} \to \frac{C_\star}{M + j_T\sigma^2}.$$

Summing this identity from $t = 1$ to $T$, we get

$$\frac{1}{\Phi_T} = \frac{1}{\Phi_0} + \sum_{t=1}^{T}\left(\frac{1}{\Phi_t} - \frac{1}{\Phi_{t-1}}\right).$$

Dividing by $T$ and applying Cesaro's theorem gives

$$\frac{1}{T\Phi_T} = \frac{1}{T\Phi_0} + \frac{1}{T}\sum_{t=1}^{T}\left(\frac{1}{\Phi_t} - \frac{1}{\Phi_{t-1}}\right) \to \frac{C_\star}{M + j_T\sigma^2}.$$

Therefore,

$$\Phi_T \sim \frac{M + j_T\sigma^2}{T\left(\sum_{j=1}^{j_T} f_j\sqrt{\tilde{\gamma}_j^T}\right)^2}. \tag{55}$$

We next consider the directions $j > j_T$. Under the optimal adversarial response in Proposition 5.2, no attack budget is allocated to these inactive directions. The defense therefore uses

$$\lambda_j^t = \frac{\sigma^2}{n_t \mathcal{R}_j^{t-1}}.$$

For such a direction, the scalar recursion without adversarial noise is

$$\mathcal{R}_j^t = (c_j^t)^2 \mathcal{R}_j^{t-1} + \frac{(1-c_j^t)^2 \sigma^2}{\tilde{\gamma}_j^t}.$$

Using

$$c_j^t = \frac{\lambda_j^t}{\lambda_j^t + \gamma_j^t} = \frac{\sigma^2}{\sigma^2 + \tilde{\gamma}_j^t \mathcal{R}_j^{t-1}},$$

we obtain

$$1 - c_j^t = \frac{\tilde{\gamma}_j^t \mathcal{R}_j^{t-1}}{\sigma^2 + \tilde{\gamma}_j^t \mathcal{R}_j^{t-1}}.$$

Substituting these two expressions gives

$$
\begin{aligned}
\mathcal{R}_j^t &= \left( \frac{\sigma^2}{\sigma^2 + \tilde{\gamma}_j^t \mathcal{R}_j^{t-1}} \right)^2 \mathcal{R}_j^{t-1} + \left( \frac{\tilde{\gamma}_j^t \mathcal{R}_j^{t-1}}{\sigma^2 + \tilde{\gamma}_j^t \mathcal{R}_j^{t-1}} \right)^2 \frac{\sigma^2}{\tilde{\gamma}_j^t} \\
&= \frac{\sigma^4 \mathcal{R}_j^{t-1}}{(\sigma^2 + \tilde{\gamma}_j^t \mathcal{R}_j^{t-1})^2} + \frac{\sigma^2 \tilde{\gamma}_j^t (\mathcal{R}_j^{t-1})^2}{(\sigma^2 + \tilde{\gamma}_j^t \mathcal{R}_j^{t-1})^2} \\
&= \frac{\sigma^2 \mathcal{R}_j^{t-1} (\sigma^2 + \tilde{\gamma}_j^t \mathcal{R}_j^{t-1})}{(\sigma^2 + \tilde{\gamma}_j^t \mathcal{R}_j^{t-1})^2} \\
&= \frac{\sigma^2 \mathcal{R}_j^{t-1}}{\sigma^2 + \tilde{\gamma}_j^t \mathcal{R}_j^{t-1}}.
\end{aligned}
$$

Taking reciprocals yields

$$\frac{1}{\mathcal{R}_j^t} = \frac{1}{\mathcal{R}_j^{t-1}} + \frac{\tilde{\gamma}_j^t}{\sigma^2}.$$

Unrolling this recursion gives

$$\frac{1}{\mathcal{R}_j^T} = \frac{1}{\mathcal{R}_j^0} + \sum_{t=1}^{T} \frac{\tilde{\gamma}_j^t}{\sigma^2}.$$

Since $\mathcal{R}_j^0 \le W$, we have $1/\mathcal{R}_j^0 \ge 1/W$, and thus

$$\mathcal{R}_j^T \le \frac{\sigma^2}{\sigma^2/W + \sum_{t=1}^{T} \tilde{\gamma}_j^t}.$$

Combining this bound over all inactive directions with (55) gives

$$\mathcal{R}(w_T) \lesssim \frac{M + j_T \sigma^2}{T \left( \sum_{j=1}^{j_T} f_j \sqrt{\tilde{\gamma}_j^T} \right)^2} + \sum_{j=j_T+1}^{p} \frac{\sigma^2}{\sigma^2/W + \sum_{t=1}^{T} \tilde{\gamma}_j^t}. \tag{56}$$

This proves the claimed asymptotic bound for our robust feature defense.

Finally, we compare with the baseline regularizer $H_t$ in (9). Under this baseline, the eigenvalue in direction $j$ is

$$\lambda_j^t = \frac{\sigma^2/W + \sum_{\tau=1}^{t-1} \tilde{\gamma}_j^\tau}{n_t}.$$

Let

$$A_{j,T} := \sigma^2/W + \sum_{t=1}^{T} \tilde{\gamma}_j^t.$$

Unrolling the corresponding linear recursion in direction $j$ gives the final estimation error as a cumulative weighted sum of the initial error, the inherent noise, and the adversarial noise. The coefficient multiplying the task-$t$ noise in direction $j$ is $\sqrt{\tilde{\gamma}_j^t}/A_{j,T}$. Therefore, if the adversary allocates variance $M$ to direction $j$ at each task, the final attack-induced variance in this direction is

$$M \sum_{t=1}^{T} \frac{\tilde{\gamma}_j^t}{A_{j,T}^2} = M \frac{\sum_{t=1}^{T} \tilde{\gamma}_j^t}{\left(\sigma^2/W + \sum_{t=1}^{T} \tilde{\gamma}_j^t\right)^2}.$$

For the stationary-maximizer instance considered in the theorem, the adversary selects the direction maximizing this quantity, so the attack-induced term is

$$M \max_j \frac{\sum_{t=1}^{T} \tilde{\gamma}_j^t}{\left(\sigma^2/W + \sum_{t=1}^{T} \tilde{\gamma}_j^t\right)^2} \sim \max_j \frac{M}{\sum_{t=1}^{T} \tilde{\gamma}_j^t}.$$

Under the stationary setting $\sum_{t=1}^{T} \tilde{\gamma}_j^t \sim T\tilde{\gamma}_j^T$, this becomes

$$\max_j \frac{M}{T\tilde{\gamma}_j^T}.$$

The benign noise contribution under $H_t$ is the standard attack-free term

$$\sum_{j=1}^{p} \frac{\sigma^2}{\sigma^2/W + \sum_{t=1}^{T} \tilde{\gamma}_j^t}.$$

Combining the attack-induced term and the benign noise term gives

$$\mathcal{R}(w_T) \lesssim \max_j \frac{M}{T\tilde{\gamma}_j^T} + \sum_{j=1}^{p} \frac{\sigma^2}{\sigma^2/W + \sum_{t=1}^{T} \tilde{\gamma}_j^t},$$

which proves the comparison in (17).

## E. Experiments

### E.1. Implementation Details

**Datasets and task construction.** All experiments are implemented in PyTorch. For both CIFAR-10 and CIFAR-100, the 50,000 training examples are randomly permuted once and split into 100 equal continual-learning tasks, so each task contains 500 training examples. After each task, we evaluate the current model on the full clean test set. Unless otherwise stated, all random seeds are fixed across compared methods so that the task order and attacked tasks are identical within each experiment.

**CIFAR-100 with frozen ViT features.** For the CIFAR-100 experiments, we first extract image features offline using a pretrained ViT-B/16 model (`vit_base_patch16_224`) from `timm` (Dosovitskiy et al., 2021). Images are resized to $224 \times 224$ and normalized with ImageNet statistics. We remove the classification head and use the pre-logit representation, giving a 768-dimensional feature vector for each image. The ViT encoder is frozen in all subsequent CL runs, and only a linear classifier from 768 dimensions to 100 classes is trained.

The linear head is trained with cross-entropy loss using AdamW with learning rate $10^{-4}$, batch size 128, and 10 epochs per task in all CIFAR-100 experiments. Since the ViT backbone is frozen, we treat the frozen feature vectors as the task inputs $X_t$ in the linear model used by our defense. For the shifted-attack experiments on CIFAR-100, the adversary poisons 10 out of the 100 tasks. The attacked tasks are sampled once and held fixed across all methods on the same dataset. On each attacked task, the adversary adds a constant shift of 10 to every coordinate of the training feature vectors. For the non-shifted attack experiments, the adversary is allowed to attack every task. We consider a challenging bounded attack with budget $M = 2000$, where the perturbation is computed by solving the adversary's inner maximization problem in (11).

**CIFAR-10 with a CNN trained from scratch.** For CIFAR-10, we train a convolutional network from scratch. The training transform uses random horizontal flips, random $32 \times 32$ crops with padding 4, conversion to tensors, and normalization by mean and standard deviation $(0.5, 0.5, 0.5)$. The test transform uses the same normalization without data augmentation. The CNN has three convolutional stages with channels $3 \to 32 \to 64 \to 128$. Each stage consists of a $3 \times 3$ convolution, ReLU, and $2 \times 2$ max pooling. The classifier contains a 256-unit hidden layer, ReLU, dropout with rate 0.2, and a final linear layer. We train with Adam, learning rate $2 \times 10^{-3}$, batch size 128, 10 epochs per task.

For the shifted, infrequent-attack experiments, the adversary poisons 10 out of the 100 tasks. In the T2T verification defense, we use a diagonal Hessian approximation in place of the exact Hessian required to compute the detection score. A task pair is flagged if the current score is larger than 2.5 times the mean score over the previous five available scores. When a task pair is flagged, we roll back the model and EWC statistics to the checkpoint before the current task and discard the poisoned update, matching the rejection step in Alg. 1.

**Memorization baseline.** For the iCaRL-style memorization baseline, we maintain a fixed exemplar memory per class and update it after each task by herding in the current feature space. For CIFAR-100 we keep 5 exemplars per class; for CIFAR-10 we keep 10 exemplars per class. The model is trained on the union of the current task and the replay memory, without the EWC penalty, so the comparison isolates the effect of memorization-based replay.

**Synthetic linear experiment.** For the synthetic experiment in Fig. 4, we use dimension $p = 8$, horizon $T = 10$, $n_t = 20$ samples per task, noise variance $\sigma^2 = 1$, attack budget $M = 10$, and initial variance $W = 1$. Each task matrix is generated with imbalanced singular values: the largest scale is 10, the smallest is 0.1, and the remaining scales are sampled uniformly from $[1, 3]$, followed by random orthogonal rotations. For the general full-matrix case, $H_t$ is optimized as $LL^\top + \epsilon I$ by Nelder-Mead from multiple random initializations. We then run 500 Monte Carlo simulations in which the adversary samples $\eta_t = \sqrt{M} z v_1$, where $z \sim \mathcal{N}(0, 1)$ and $v_1$ is the top eigenvector of the corresponding sensitivity matrix. The figure below reports the mean squared parameter error for our defense and for the EWC regularizer in (9).

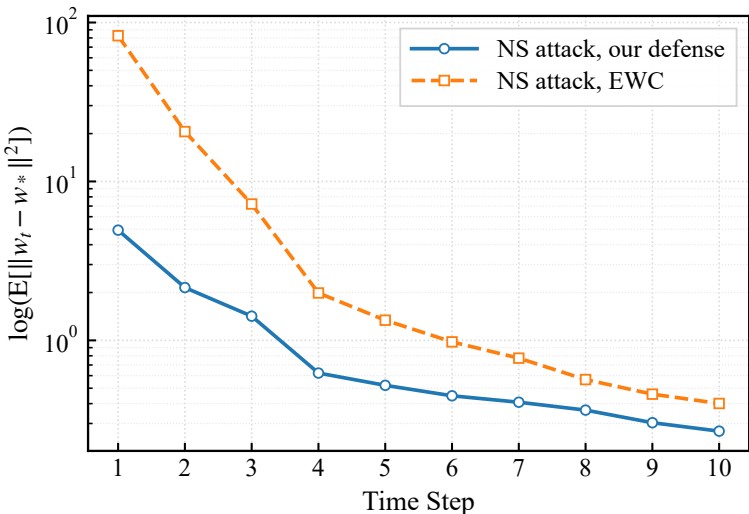

*Figure 4.* Log excess risk $\mathcal{R}(w_t)$ under our robust defense and EWC in (9) against strategic attack.

