# OpenReview forum: "Theory of Continual Learning Against Data Poisoning Attacks"
_ICML.cc/2026/Conference — ICML 2026 regular_

### Official Review · Reviewer_juUT · 2026-03-05

**Soundness:** 4
**Presentation:** 4
**Significance:** 3
**Originality:** 3
**Overall Recommendation:** 5
**Confidence:** 2

**Summary:**

This paper develops a framework for adversarial learning in the context of continual learning.
Specifically, the authors focus on attacks and defenses in regularization-based continual learning.
In addition, the paper presents an empirical investigation on CIFAR data, supporting the provided theory.

**Compliance With Llm Reviewing Policy:**

Affirmed.

**Key Questions For Authors:**

1. Regarding the threat model: The attacker knows the currentl CL model state and dataset (as stated in 2.2). Do they, however, know (anything about) the future tasks/datasets?
And if they do(/don't), does the theory change substantially if they didn't(/did)?

2. In section 5, paragraph 1: "Since these attacks are bounded and do not induce a distributional shift, the poisoning term eta tidle t (3) can be interpreted as an augmented form of inherent noise to some extent"
Can you elaborate on that?

3. The attacker (in the bounded case) is bound by the (expected) Frobenius norm of the perturbation (as stated just above eqn (3)). How much of the analysis and theory (if any) carries through in cases where the attack is bound by some other function? Canonical examples that come to mind are l1 and l-infinity norms.

**Limitations:**

yes

**Strengths And Weaknesses:**

Strengths:

- The paper presents rigorous, theoretical results that are useful while also providing intuition.

- For some attacks, a justified defense is also proposed.

- The paper is well written and easy to follow.

Weaknesses:

- Experiments are performed only on vision tasks.

- The only bounding function for the attacker considered is Frobenius norm.

Overall:

This paper addresses a gap in the adversarial literature. Namely, it considers continual learning and forms a theoretical foundation for CL in adversarial context. The main finding is intuitive: attackers "win" when they have the ability to add unbounded noise to a large (linear) proportion of the data. In addition to the formalization and proof of this main result, the authors address the settings where defenses may actually succeed (bounded and infrequent attacks). This provides a foundation by which the community can better understand adversarial CL settings. In addition, the authors propose several defenses for the various settings. In total, this paper provides a service to both researchers (forming the foundational base of understanding) and practitioners (presenting deployable defenses).

---

> ### Author Rebuttal · Authors · 2026-03-31
>
> We sincerely appreciate your time and thoughtful feedback. Below is our detailed response.
>
> 1.Knowledge of future tasks by the attacker: Usually, the learner has no less information than the attacker in the defense problem, since the learner also has access to its own task data for learning. Here we assume the chanllenging case that the attacker has the same information as the defender. Therefore, the attacker does not know future tasks or future datasets, just as the defender does not. If the attacker were granted access to future information, it would become unrealistically powerful, as it could optimize its attack globally over all future CL update states and arbitrarily mislead the system throughout the entire learning process. In that case, effective defense may become impossible, consistent with our impossibility results in Section 3.
>
> 2.Interpreted as an augmented form of inherent noise: After further consideration, we believe this statement may be somewhat misleading and would like to revise it. The noise injection formulated in Eq. (11) represents a strategic attack mechanism, which is fundamentally different from the more common setting where the attack is modeled as uniform noise. Please also refer to our Response 2 to Reviewer LCJK for a more detailed discussion of the distinction between strategic noise injection and uniform noise.
>
> 3.Experiments on vision tasks: Thank you for the comment. We focus on CV tasks primarily because the most widely used and well-established CL benchmarks focus on in the CV domain, which has made vision the standard testbed for evaluating new CL methods. In addition, many current CL works also validate their methods mainly on vision tasks, e.g., "Deng, Junze, et al. (ICML 2025)" and "Li, Hongbo, et al. (ICLR 2025)". Therefore, our experimental scale and setup follow the standard evaluation protocol commonly adopted in the literature. Still, our experiments can be readily extended to non-computer vision tasks, such as time-series regression, should that be of interest.
>
> 4.Attack with Frobenius norm: We can also handle budget constraints under other norms. We can replace the Frobenius norm constraints in Eqs. (5) and (11) with any other norm constraints, and the defender can still solve the optimization-based defense problem in Eqs. (5) and (11) to design the regularizer optimally, so our attack-defense framework remains valid.
>
> Regarding the detailed optimal solutions to Eqs. (5) and (11) under other norm constraints, they are expected to be similar to our current theoretical results under the Frobenius norm. For example, the induced matrix norm $\\|\cdot\\|_1$ satisfies $\\|\cdot\\|_1/\sqrt{d} \le \\|\cdot\\|_F \le \sqrt{d}\\|\cdot\\|_1$. This implies that the optimal solution to Eqs. (5) and (11) under Frobenius norm remains feasible under constraints formulated with the $\\|\cdot\\|_1$ norm. Moreover, since $\\|\cdot\\|_1$ norm also captures imbalance among the entries of a matrix, replacing the Frobenius-norm constraints in Eqs. (5) and (11) with $\\|\cdot\\|_1$ norm leaves the core idea of our theoretical results unchanged. The extension for $\\|\cdot\\|\_{\infty}$ induced matrix norm holds similarly.

---

> > ### Author Rebuttal · Reviewer_juUT · 2026-04-03
> >
> > Thanks!

---

> > > ### Author Response · Authors · 2026-04-07
> > >
> > > Thank you very much for your careful review and for acknowledging our work.

---

### Official Review · Reviewer_LCJK · 2026-03-10

**Soundness:** 3
**Presentation:** 4
**Significance:** 2
**Originality:** 3
**Overall Recommendation:** 4
**Confidence:** 2

**Summary:**

This paper investigates robust learning against data poisoning attacks in the continual learning setting. In this setting, the adversary wants to harm the convergence rates within some attack budget. They study the ability of regularization-based algorithms against different types of adversaries. They show that when the adversary has frequent unbounded attacks or frequent shifted attacks, it is not defensible. Then, when the attack is unbounded/shifted but infrequent, they design an algorithm called Task-to-task verification to defend against those attacks. And when the attacks are frequent but bounded, they use the robust feature defense to defend against the adversary. Both of the methods are analyzed theoretically and experimentally.

**Compliance With Llm Reviewing Policy:**

Affirmed.

**Final Justification:**

I thank the authors for their detailed responses.

**Key Questions For Authors:**

1. In this paper, the goal of the adversary is to slow down the convergence by providing poisoned data. Is it possible to change the goal to mislead the model on some specific data point or some specific task? What would be the defensibility under this goal? I think this may be a more interesting question than discussing the harm to the convergence as a whole.
2. What about other learning methods? Is it possible for them to defend against worse adversaries than those regularization-based algorithms?
3. Is it possible to view the case of frequent bounded attacks as a continual learning with noise? This will make the problem more similar to the agnostic learning in learning theory, which may give some more general results.

**Limitations:**

Yes

**Strengths And Weaknesses:**

The paper is well written. The structure of the paper is very clear and easy to follow. The problem itself is important and interesting. I agree that nowadays, when learning models are ubiquitous, the output is heavily related to the training data. The volume of data required when training a modern learning model is very huge, and it is very hard to clean all those data manually; therefore, it is reasonable to design an adversarially robust learning model that is robust against those poisoning data attacks. However, the model discussed in this paper is very simple, and the results do not seem to be surprising. The design of the algorithm also seems similar to the one in Awasthi et al. 2017, "The Power of Localization for Efficiently Learning Linear Separators with Noise". When the attacks are unbounded but infrequent, detect them and just drop the update of that task. When the attack is frequent but bounded, treat this problem as learning with bounded noise. It feels that the theoretical results are not very novel or strong.

---

> ### Author Rebuttal · Authors · 2026-03-31
>
> We sincerely appreciate your time and thoughtful feedback. Below is our detailed response.
>
> 1.Theoretical novalty: We agree that our T2T defense is somewhat related to the outlier-related method of Awasthi et al. (2017). However, their approach applies only to the linear separator setting and cannot be used for our continual learning problem with linear regression, which is more general. Moreover, as shown in Eqs. (5) and (11), our framework extends to neural network training in nonlinear settings, whereas their method is limited to linear separators and cannot be adapted to our problem.
>
> Further, they only consider the attack in low frequency, while, in our Section 5, we consider a more challenging setting in which the adversary can attack every task and continuously inject poisoning noise. Their defense framework is not applicable in this regime. In particular, our setting requires continual learning techniques that remain effective under persistent poisoning, whereas the soft outlier technique in their method would fail when every task is poisoned.
>
> 2.Strategic attack, not mere noise: The attack model we study is fundamentally different from bounded noise. In continual learning, uniformly bounded noise is comparatively easy to defend against. As shown in our additional experiments below, the standard CL method EWC still achieves a final test accuracy of 0.59 under uniform noise, whereas the strategic noise injection considered in our work leads to a severe performance drop:
> $$
> \\begin{array}{l|cccccccc}
> \\hline
> \\mathrm{Test\\ accuracy\\ on\\ CIFAR-10 (\\%)}
> & t=10 & t=20 & t=30 & t=40 & t=50 & t=60 & t=70 \\\\
> \\hline
> \\text{Uniform attack }
> & 48 & 50 & 52 & 55 & 57 & 57 & 59  \\\\
> \\text{Strategic attack}
> & 23 & 33 & 40 & 40 & 44 & 47 & 48 \\\\
> \\hline
> \\end{array}
> $$
> In our attack model, as formulated in Eqs. (5) and (11), the adversary acts strategically to maximize model damage. As discussed from line 334 in the left column of page 7, when maximizing the objective in Eq. (11), the adversary concentrates its poisoning budget along the feature eigendirection that is most sensitive to attack, rather than spreading the perturbation uniformly across the dataset. This makes the attack substantially more harmful and much harder to defend against than ordinary noise. Consequently, modeling the attack merely as standard bounded noise fails to capture the minimax interaction between the attacker and defender in Eq. (11). This is why a new defense, such as that proposed in Proposition 5.2, is needed. We therefore believe our theoretical results are novel in addressing this new class of attacks.
>
> 3.Simplicity of our model: The theoretical study of continual learning is still a new area at an early stage, and as a result, most existing theoretical works focus on relatively simple models. For example, Deng, Junze, et al. (ICML 2025), Li, Hongbo, et al. (ICLR 2025), and Peng, Liangzu, et al. (ICLR 2025) all develop their analyses under linear models. In comparison, our work already extends the theoretical formulation from the linear setting to the nonlinear setting, as reflected in Eqs. (5) and (11), and is applicable for non-linear seting both theoretically and experimentally, thereby relaxing this limitation.
>
> 4.Attack to mislead the model on specific data points: Thank you for the valuable comment. We would like to clarify that the primary purpose of our attacker model is not merely to slow convergence, but to increase the CL's generalization loss. Because our formulation explicitly models the attacker's maximization problem, any alternative attack objective cannot be more damaging than the worst-case attack considered in our framework with respect to the defender's generalization-loss objective. Therefore, the theoretical upper bounds on generalization loss in Theorems 4.4 and 5.3 also apply to other attack objectives.
>
> Moreover, the formulations in Eqs. (5) and (11) can be adapted to capture alternative attacker goals, such as misleading the model on specific data points. In such cases, the defender can still solve the corresponding optimization problem to design an appropriate defense.
>
> 5.Other learning method: Since we study a minimax formulation within the regularization-based continual learning framework, our method optimizes the regularizer against the worst-case attack. Therefore, it is already optimal within the class of regularization-based CL methods, and no other method in this class can be expected to outperform it under the same formulation. For other non-regularization-based continual learning algorithms, most are heuristic methods designed for specific attack models and do not provide robust guarantees of successful protection.
>
> If our clarifications and additional results have addressed your concerns, we would greatly appreciate your consideration in updating your rating. We would also be happy to answer any further questions.

---

> > ### Author Rebuttal · Reviewer_LCJK · 2026-04-02
> >
> > I thank the authors for their detailed response. My questions have been fully resolved, and I will raise my score accordingly.

---

> > > ### Author Response · Authors · 2026-04-07
> > >
> > > Thank you very much for your careful review and for acknowledging our work.

---

### Official Review · Reviewer_ZYci · 2026-03-11

**Soundness:** 3
**Presentation:** 2
**Significance:** 3
**Originality:** 3
**Overall Recommendation:** 4
**Confidence:** 3

**Summary:**

This paper focuses on the robustness of Regularization-based Continual Learning against data poisoning, proposing a theoretical framework and defense strategies. Specifically, the authors model the interaction between attacker and defender as an online zero-sum game and prove a fundamental performance limit: when an adversary poisons a linear proportion of tasks by adding unbounded noise or pattern shifts, any defense fails. Based on this finding, two potentially defensible scenarios are proposed, along with corresponding countermeasures.

**Compliance With Llm Reviewing Policy:**

Affirmed.

**Final Justification:**

The authors effectively resolved my confusion; I am maintaining a positive rating.

**Key Questions For Authors:**

1. Are the theoretical assumptions mild?

2. While some approximations can be used to approximate the Hessian matrix and its inverse, what are the typical time and space costs of such approximations, and what is their impact on defense?

**Limitations:**

Please refer to weaknesses.

**Strengths And Weaknesses:**

Strengths:
1. A theoretical framework for CL robustness against data poisoning is proposed, proving that if an attacker contaminates a linear proportion of tasks with unbounded noise or data pattern shifts, a provable convergence guarantee cannot be established.

2. A task-to-task verification mechanism is proposed.

3. A robust feature defense against frequent, bounded, and non-shifted attacks is derived.

4. CIFAR experiments clearly demonstrate that the proposed algorithm provides a metric that effectively reveals attack threats.

Weaknesses:

1. My main concern is the reasonableness of the theoretical assumptions. The proposed theory heavily relies on assumptions of commutativity, linear models, etc., which seem difficult to satisfy in practice, as networks are typically highly nonlinear neural networks. While I understand the difficulty of analyzing highly complex networks, the paper needs more discussion on the reasonableness of the assumptions.

2. The overall computational complexity of the proposed algorithm needs to be verified both theoretically and empirically.

3. The experimental setup is unclear. The paper lacks details on the execution of the attack, the attacker's capabilities, and budget.

---

> ### Author Rebuttal · Authors · 2026-03-31
>
> We sincerely appreciate your time and thoughtful feedback. Below is our detailed response.
>
> 1.Relaxition of Assumption 5.1 on commutativity: We would like to clarify that Assumption 5.1 is introduced mainly to obtain a closed-form solution in Proposition 5.2 as well as the closed-form convergence rate in Theorem 5.3 to provide insights. We relax Assumption 5.1 below by optimizing $H_t$ from the following generalized optimization problem from problem (11):
> $$\min_{H_t}\max_{\mathrm{Cov}(\tilde{\eta}_t)}\bigg\\|P_t\bigg(H_t \mathbb{E}[w\_{t-1}- w\_*]\bigg)\bigg\\|^2+\mathrm{tr}\left(P_t\frac{X_t^\top}{n_t}\mathrm{Cov}(\tilde{\eta}_t)\frac{X_t}{n_t}P_t\right) +\mathrm{tr}\left(\sigma^2P_t\frac{X_t^\top X_t}{n_t^2}P_t\right)+\mathrm{tr}\left(P_t H_t^\top C_t H_tP_t\right)$$
>
> subject to $\\mathrm{tr}\\big(\\mathrm{Cov}(\\tilde{\\eta}_t)\\big)\\leq M$, where
>
> $P_t=(\frac{X_t^\top X_t}{n_t}+H_t)^{-1}$,
>
> $C_t=\mathrm{Cov}(w_{t-1}-w_{\ast})$,
>
> and $\tilde{\eta}_t$ denotes the adversarial noise.
>
> The corresponding recursions for the mean $\\mathbb{E}[w_t-w_{\\ast}]$ and covariance $C_t$ under $H_t$ and $\\tilde{\\eta}_t$ are computed below:
>
> $$\\mathbb{E}[w_t-w_*]=P_tH_t\\mathbb{E}[w_{t-1}-w_*],$$
>
> $$C_t=P_t\frac{X_t^\top}{n_t}\mathrm{Cov}(\tilde{\eta}\_t)\frac{X_t}{n_t}P_t+P_tH_t^\top C_{t-1}H_tP_t+\sigma^2P_t\frac{X_t^\top X_t}{n_t^2}P_t.$$
>
> Therefore, to implement the defense, the system can recursively solve the mean and covariance for $H_t$ using the optimization problem above. This is an optimal defense for $H_t$ in the regularization-based framework against the worst-case attack and is strictly better than the other regularization method.
>
> We have added the following experiments, which do not rely on Assumption 5.1, to illustrate the empirical convergence behavior as supplementary evidence for Theorem 5.3. At each step, we construct an $20\times 8$-dimensional feature matrix $X_t$ with imbalanced singular values to simulate the attack-defense process, where the adversary chooses the optimal $\tilde{\eta}_t$ against both our optimal $H_t$ and the EWC benchmark. The resulting generalization loss shows that our method drives the loss to zero much faster, aligning with Theorem 5.3.
> $$\\begin{array}{l|cccccccc}\\hline\\mathrm{Generalization\\ loss}&t=1&t=2&t=3&t=4&t=5&t=6&t=7\\\\\\hline H_t \\mathrm{\\ solved\\ from\\ problem\\ above}
> &5.015921&2.228334&1.419592&0.643991&0.524078&0.457405&0.410549\\\\\\text{Benchmark }H_t\\text{ in Eq.(9)}&83.325343&20.222377&7.141862&1.998422&1.371249&0.983108& 0.773966\\\\\\hline\\end{array}$$
> $H_t$ can be computed by using a diagonal regularization, which has complexity $O(d)$. Our method can also be extended to nonlinear model by our formulations in Eqs. (5) and (11). Please refer to our Fig. 2(b) and Response 3 to Reviewer XigQ.
>
> 2.Computational complexity: In practice, our T2T defense against the attack in Section 4 requires the Hessian and its inverse, both of which can be approximated using standard methods rather than computed exactly. For example, a diagonal Hessian only computes diagonal second-order derivatives, with computation, storage, and inversion cost of $O(d)$, the parameter dimension. Similarly, Gauss-Newton approximation requires $O(vd)$ storage, and $O(v^2 d)$ in inverse computation by Woodbury identity, where the output dimension $v$ is typically much smaller than $d$.
>
> In our T2T simulations including Fig. 2, we already use the diagonal approximation to compute the detection score in Eq. (6). Its storage cost is small, at about 2MB. Fig. 2 shows that the detection score successfully identifies attacks under this approximation, confirming that the method remains effective with approximate Hessian.
>
> For the robust feature defense against frequent and bounded attack in Section 5, we start from the Fisher information regularizer and then apply additional gradient descent on it to solve Eq. (11) with similar complexity $O(d)$.
>
> 3.Experimental setup: For simulating CL for CIFAR-10 dataset, we use convolutional neural network with three convolutional stages ($3\rightarrow 32\rightarrow 64\rightarrow 128$). Each stage includes a $3\times 3$ convolution, ReLU, and a $2\times 2$ max-pooling, followed by a fully connected layer with $256$ hidden units and a final linear layer for prediction. For the shifted attack on CIFAR-10, we consider a challenging unbounded attack with large budget $3072\times2$, corresponding to adding a shift of 2 to every input pixel. For CIFAR-100, the attacker adds $768\times50$ to every output coordinate of the frozen feature layer. For the non-shifted attack, the adversary injects high-variance noise (variance 20) into the most attack-sensitive one-hot label coordinates, selected according to Proposition 5.2. We will add such details.
>
> If our clarifications and additional results have addressed your concerns, we would greatly appreciate your consideration in updating your rating. We would also be happy to answer any further questions.

---

> > ### Author Rebuttal · Reviewer_ZYci · 2026-04-01
> >
> > I thank the authors for effectively resolving my concerns; I have no further questions. I will maintain a positive rating.

---

> > > ### Author Response · Authors · 2026-04-07
> > >
> > > Thank you very much for your careful review and for acknowledging our work.

---

### Official Review · Reviewer_XigQ · 2026-03-13

**Soundness:** 3
**Presentation:** 2
**Significance:** 3
**Originality:** 3
**Overall Recommendation:** 4
**Confidence:** 3

**Summary:**

This paper studies data poisoning in regularization-based continual learning (CL), framing the attacker-defender dynamics as an online zero-sum game. The authors categorize attacks by frequency, magnitude, and pattern, and present three main results. First, they show an impossibility: if the adversary poisons a linear fraction of tasks with unbounded noise or distributional shifts, no defense can guarantee convergence (Thm 3.1). Second, for infrequent but unbounded attacks, they propose a Task-to-Task (T2T) verification scheme that detects poisoned tasks and achieves O(K²/T) convergence (Thm 4.4). Third, for frequent, bounded, non-shifted attacks, they solve for optimal regularization via Nash equilibrium to speed up convergence (Prop 5.2, Thm 5.3). Experiments on CIFAR-10/100 are provided.

**Compliance With Llm Reviewing Policy:**

Affirmed.

**Final Justification:**

The author's clarifications and additional results have partially addressed my concerns, I'd like to update my rating.

**Key Questions For Authors:**

1. Where are the non-linear results for the robust feature defense? This is a pretty central piece of the paper and it’s just missing.
2. How sensitive is T2T detection to getting σ² wrong? In practice you’d have to estimate it — does detection still work reasonably well with a noisy estimate?
3. Is there any hope of relaxing Assumption 5.1? Even approximate commutativity or block-diagonal structure would make the results much more applicable.
4. The O(K²/T) bound in Thm 4.4 — is the quadratic dependence on K tight, or an artifact of the proof? It’d be nice to know if O(K/T) is achievable.
5. Only two baselines seems insufficient. Could you add comparisons with more recent CL methods or standard poisoning defenses?

**Limitations:**

The authors do mention the missing non-linear experiments and the strong assumptions. But they don’t discuss the gap between the theoretical threshold θₜ and the rolling-window heuristic actually used in Alg. 1, which I think is worth flagging.

**Strengths And Weaknesses:**

**Strengths**
1. I like the way the paper organizes the attack space into three dimensions (frequency, magnitude, pattern) — Table 1 gives a nice overview and makes it easy to see which regimes are defensible and which are not. This taxonomy feels useful beyond just this paper.
2. The theoretical results are solid. The impossibility result (Thm 3.1) is clean and informative. The T2T detection score (Eq. 6–8, Lemma 4.1) is a clever idea — decoupling historical drift from local task dynamics is a neat trick. The Nash equilibrium derivation for the regularizer (Prop 5.2) is also well done.
3. This fills a real gap. There’s been plenty of empirical work on adversarial CL, but almost no rigorous theory. This paper is among the first to give formal treatment of attack and defense in regularization-based CL, which I think is valuable.

**Weaknesses**
1. The assumptions are quite strong and not adequately discussed. Assumption 5.1 requires the Hessians {X⊤ₛXₛ} to commute, which is a big ask in practice. Thm 5.3 also needs the protected set {1,...,jₜ} to stay the same across tasks. I'd really like to see some discussion of what happens when these don't hold exactly — do the results degrade gracefully, or do they break down?
2. The experiments for the robust feature defense (Section 5) are incomplete. The authors themselves say they're "currently conducting" the non-linear simulations. This is a significant omission — you can't really claim the defense works in practice without showing it. For an ICML submission, I'd expect all experiments to be done.
3. Only EWC and iCaRL as baselines feels thin. There are more recent CL methods and also dedicated data poisoning defenses that should be compared against. As it stands, it's hard to tell how much the proposed approach actually improves over the broader landscape of methods.
4. The theoretical threshold θₜ for T2T detection (Lemma 4.3) requires knowing σ², which you typically don't have. The actual algorithm (Alg. 1, Line 8) uses a rolling-window heuristic instead, but there's no guarantee given for that version. So there's a gap between what's proved and what's actually used.
5. There are quite a few typos scattered throughout the paper, e.g., "denfensible" in the abstract, "magnitute" in Section 2, "moveing" in Section 5. These are just a few I noticed; there are probably more. A careful proofreading pass is needed.

---

> ### Author Rebuttal · Authors · 2026-03-31
>
> We sincerely appreciate your time and thoughtful feedback. Below is our detailed response.
>
> 1.Relaxition of Assumption 5.1 on commutativity: The optimality and implementation of our robust feature defense does not rely on Assumption 5.1 and the assumption in Theorem 5.3. Please refer to our Response 1 to Reviewer ZYci for more details on its relaxition.
>
> 2.Noise estimation for T2T defense: We can estimate an upper bound $\bar{\sigma}^2$ and use it in the T2T defense in the linear case with unknown noise $\sigma^2$. In practice, estimating such an upper bound is straightforward, since one can always choose $\bar{\sigma}^2$ large enough to ensure that it is valid. In this case, the generalization loss upper bound in Theorem 4.4 becomes linear in $\bar{\sigma}^2$ instead of $\sigma^2$, while preserving the same convergence order of $O(K^2/T)$.
>
> For nonlinear setting, we are inspired by results in Lemma 4.3 and Theorem 4.4 to propose the heuristic rolling-window method. Theoretical analysis for nonconvex deep neural networks remains highly challenging with all the current techniques, and it is widely accepted to focus on theoretical analysis on linear model to inspire the design for nonlinear models. The following works follow a similar paradigm: “Unlocking the Power of Rehearsal in Continual Learning (ICML 2025)”, “Theory on Mixture-of-Experts in Continual Learning (ICLR 2025)”, "Theory on Forgetting and Generalization of Continual Learning (ICML 2023)"
>
> 3.Experiments for nonlinear model: We have now completed all experiments, including the nonlinear setting in Section 5 on the CIFAR-10 dataset using a convolutional neural network. To simulate the strategic attack in the nonlinear model, the adversary computes the gradient of the current loss with respect to each input element and selects the top $16\%$ of the 3072 CIFAR-10 image pixels to receive the poisoning budget. For defense, we use a diagonal Hessian approximation to compute the optimal $H_t$. The test accuracies of our method, EWC, and iCaRL are shown below:
> $$
> \\begin{array}{l|cccccccc}
> \\hline
> \\mathrm{Test\\ accuracy} (\\%)
> &  t=30 & t=40 & t=50 & t=60 & t=70 & t=80 & t=90 \\\\
> \\hline
> \\text{Robust feature defense}
> & 34.85 & 39.25 & 35.41 & 39.06 & 34.52 & 40.81 & 50.12  \\\\
> \\text{EWC}
> & 39.61 & 34.16 & 36.56 & 38.40 & 38.35 & 39.92 & 37.88   \\\\
> \\text{iCaRL}
> & 39.91 & 36.11 & 38.56 & 36.72 & 36.68 & 35.04 & 44.49 \\\\
> \\hline
> \\end{array}
> $$
> In the nonlinear setting, our defense may initially underperform other methods due to approximation error. However, it maintains an upward trend in test accuracy and gradually converges toward the ground-truth model, whereas the other methods may converge to an incorrect model by overfitting the noise.
>
> 4.$O(K^2/T)$ bound in Thm 4.4: We believe that the order $O(K^2/T)$ is already tight. The factor $K^2$ arises because we evaluate both the quadratic loss $\mathbb{E}[\\|w_t-w^*\\|^2]$ and deterministic attack.
>
> As such, the poisoning effects accumulate coherently in $w_t$, and the error behaves like $\mathbb{E}\left[\left\\|\sum_{k=1}^K \eta_k + w_t - w^*\right\\|^2\right]$. If the attack is simplified to be random with zero-mean, the cross terms above could cancel in expectation, leading to $O(K/T)$ dependence.
>
> 5.Typos: We thank the reviewer for pointing this out. We will carefully proofread the paper, correct these errors throughout the manuscript, fix any remaining issues, and improve the overall presentation.
>
> 6.Baseline methods: We have considered other regularization-based CL method such as Synaptic Intelligence than EWC. However, since our methods already optimally solve the design problem of regularizer against the worst case attack in Eqs. (5) and (11), other regularization-based methods will not perform better then ours in the worst case.
>
> If our clarifications and additional results have addressed your concerns, we would greatly appreciate your consideration in updating your rating. We would also be happy to answer any further questions.

---

### Decision · Program_Chairs · 2026-04-30

**Decision:**

Accept (regular)

**Comment:**

The paper develops a theoretical framework for understanding data poisoning in continual learning, organizing attacks by frequency, magnitude, and shift structure, and identifying which regimes are fundamentally defensible versus impossible to guard against. It introduces a game-theoretic attacker-defender perspective and proposes two defenses tailored to the regimes where robustness is achievable. The work is primarily theoretical and stands out for providing a structured taxonomy and principled analysis, rather than relying solely on empirical attack-defense evaluations.

From a soundness perspective, the results appear correct within their modeling assumptions, and the appendices provide supporting derivations without obvious flaws. The impossibility result (Theorem 3.1) is particularly convincing, while the later defenses rely on approximations and stronger structural assumptions, such as commutativity and stationarity conditions. The main weaknesses lie in the strength of assumptions, the gap between theory and implementation, and limited empirical evaluation. However, these do not outweigh the contribution, which provides a useful conceptual foundation for studying poisoning in continual learning. For the camera-ready version, the authors should clearly distinguish proven results from heuristic components, discuss the sensitivity to assumptions, strengthen empirical validation (including additional baselines and implementation details), and improve clarity of presentation by incorporating the rebuttal. Overall, the work is a meaningful theory contribution with promising directions for future research.